# Atomic-scale combination of germanium-zinc nanofibers for structural and electrochemical evolution

Gyujin Song [1,6], Jun Young Cheong[2,6], Chanhoon Kim[3], Langli Luo [4], Chihyun Hwang[1], Sungho Choi [5], Jaegeon Ryu[5], Sungho Kim [1], Woo-Jin Song[5], Hyun-Kon Song[1], Chongmin Wang [4], Il-Doo Kim [2] & Soojin Park[5]

Alloys are recently receiving considerable attention in the community of rechargeable batteries as possible alternatives to carbonaceous negative electrodes; however, challenges remain for the practical utilization of these materials. Herein, we report the synthesis of germanium-zinc alloy nanofibers through electrospinning and a subsequent calcination step. Evidenced by in situ transmission electron microscopy and electrochemical impedance spectroscopy characterizations, this one-dimensional design possesses unique structures. Both germanium and zinc atoms are homogenously distributed allowing for outstanding electronic conductivity and high available capacity for lithium storage. The as-prepared materials present high rate capability (capacity of ~ 50% at 20 C compared to that at 0.2 C-rate) and cycle retention (73% at 3.0 C-rate) with a retaining capacity of 546 mAh $g^{-1}$ even after 1000 cycles. When assembled in a full cell, high energy density can be maintained during 400 cycles, which indicates that the current material has the potential to be used in a large-scale energy storage system.

[1] School of Energy and Chemical Engineering, Ulsan National Institute of Science and Technology (UNIST), Ulsan 44919, Republic of Korea. [2] Department of Materials Science and Engineering, Korea Advanced Institute of Science and Technology, 291 Daehak-ro, Yuseong-gu, Daejeon 34141, Republic of Korea. [3] Clean Innovation Technology Group, Korea Institute of Industrial Technology, 102 Jejudaehak-ro, Jeju-si, Jeju-do 63243, Republic of Korea. [4] Environmental Molecular Sciences Laboratory, Pacific Northwest National Laboratory, 902 Battelle Boulevard, Richland, WA 99352, USA. [5] Department of Chemistry, Division of Advanced Materials Science, Pohang University of Science and Technology (POSTECH), Pohang 37673, Republic of Korea. [6] These authors contributed equally: Gyujin Song, Jun Young Cheong. Correspondence and requests for materials should be addressed to C.W. (email: Chongmin. Wang@pnnl.gov) or to I.-D.K. (email: idkim@kaist.ac.kr) or to S.P. (email: soojin.park@postech.ac.kr)

With the increase in energy consumption and the development of large-scale devices such as electric vehicles (EVs), the demand for rechargeable energy storage systems, especially lithium-ion batteries (LIBs), has sharply increased[1–10]. However, existing LIBs are facing some challenges related to their low energy density. So far, various anode materials possessing a higher specific capacity have been suggested to replace the most widely used graphite which has a theoretical capacity of 372 mAh g$^{-1}$. Among the many candidates, germanium (Ge) is one of the most promising ones owing to its high gravimetric and volumetric capacity (1396 mAh g$^{-1}$ and 7366 Ah L$^{-1}$, respectively, for Li$_{3.75}$Ge), which are comparable to those of Si; reasonable Li ion diffusivity and electronic conductivity[11–13]. Furthermore, Ge-based anodes operate at a low operating voltage (<0.5 V), resulting in a high potential window when assembled as full cells[14]. Nevertheless, pure Ge compounds are expensive, and the synthesis of pure Ge nanostructures is rather complicated. As a result, oxygen-containing Ge (GeO$_x$) has recently drawn attention as an alternative to pure Ge due to the following merits: a higher theoretical capacity (2152 mAh g$^{-1}$), reduced cost, and superior chemical stability[15–18]. However, GeO$_x$ also triggers a large volume change during lithiation process, resulting in fatal capacity decay upon cycling with the loss of electrical contacts, fracturing, and pulverization, along with the continuous formation of unwanted solid electrolyte interphase (SEI) layers. Additionally, the poor electronic conductivity of GeO$_x$ limits electron transfer at high current density, which remains a challenge to realizing high-power/high-energy-density anode materials for advanced LIBs.

To overcome the problems mentioned above, two main strategies have been attempted: (i) one approach is to synthesize low-dimensional Ge/GeO$_x$ nanomaterials (nanoparticles, nanotubes, nanowires, and nanofibers (NFs)), which can mitigate a large volume change of >300%, shorten the Li ion diffusion length, and suppress contact loss and unstable SEI layer formation[13,19–21]; (ii) the other approach is to combine Ge/GeO$_x$ with foreign atoms (carbon, metal, or metal oxide) and/or a protective layer (carbon), which allows additional electron pathways through enhanced conductivity[22–27]. Nevertheless, a feasible strategy to enhance both the structural and electrochemical stability of GeO$_x$-based anodes has yet to be fully investigated; such a strategy is critical to realize high-performance anodes for sustainable LIBs.

In this work, we report the defect engineering of one-dimensional Ge-based materials through the intermolecular incorporation of Zn element. The as-synthesized oxygen-defective and intermolecularly distributed Ge–Zn NFs (oxygen-deficient Ge–Zn composite NFs, denoted as O-dGZNFs), which is prepared by a facile electrospinning followed by subsequent thermal treatment, feature a well-mixed atomic distribution of Ge–Zn–O with disordered Ge. In addition, this unique structure limits the sublimation of Ge or GeO$_x$, a chronic problem in solid–gas reduction reactions[28]. In situ transmission electron microscopy/electrochemical impedance spectroscopy (TEM/EIS) characterizations further demonstrate the significantly improved structural stability and electronic conductivity. Unlike previous work on Zn$_x$Ge$_{1-x}$O composites showing inferior electrochemical performance (510 mAh g$^{-1}$ at 0.5 A g$^{-1}$)[29], O-dGZNFs electrode exhibits ultrahigh cycling stability (capacity retention of 73%), a reversible capacity of 546 mAh g$^{-1}$ for 1000 cycles at 3.0 C-rate and exceptional rate capability (capacity of ~50% at 20 C-rate compared to that of 0.2 C-rate). In a full cell, a high energy density of 335 Wh kg$^{-1}$ (565 Wh L$^{-1}$) is achieved after the 1st cycle, and a stable charge/discharge characteristic is observed with Coulombic efficiency of 99.4% during 400 cycles.

## Results

**Structural evolution of O-dGZNFs.** Unstable Ge states during the reduction process result in unexpected sublimation, causing a lower yield and structural deformation. Zn atoms that are directly interconnected to GeO$_2$ could stabilize the Ge state owing to the robust Ge–Zn bonding. Based on the binary phase diagrams of various metals, Zn was carefully selected. Some metals (such as Cu and Fe) form alloys with Ge as verified by X-ray diffraction (XRD) analysis before and after reduction (Supplementary Fig. 1a). Moreover, other kinds of metals (such as Ag, Au, Sb, and Sn) can be combined with Ge, but other limitations are present. For example, Ag and Au are very expensive, and synthetic conditions are difficult due to their sensitive precursors; Sb is difficult to mix with Ge in electrospinning solution; and Sn can be easily mixed with Ge, however, undesired phases (i.e., partial Ge oxides and Sn oxides) are independently formed, limiting the generation of complete GeSn alloy at operating temperature (Supplementary Fig. 1b). As a result, we selected Zn as an optimal element to synthesize O-dGZNFs. The synthetic process of oxygen-including Ge–Zn composite NFs (O-iGZNFs) and O-dGZNFs and their TEM images are displayed in Fig. 1a. Similarly, oxygen-including Ge NFs (O-iGNFs) and oxygen-deficient Ge NFs (O-dGNFs) (control samples without Zn) were prepared using the same process, and the elemental compositions of both O-dGNFs and O-dGZNFs along with their calcined samples were determined (Supplementary Fig. 2). Although both O-dGNFs and O-dGZNFs have a high proportion of Ge, the Ge weight loss from O-iGZNFs to O-dGZNFs is less than that from O-iGNFs to O-dGNFs. An as-prepared electrospinning solution including a metal salt precursor (zinc nitrate hexahydrate/germanium oxide) and a poly (vinyl pyrrolidone) (PVP) was injected and directly transformed into polymer/metal ion composite NFs on a current collector, as shown in the scanning electron microscopy (SEM) image in Fig. 1b. Then, upon calcination, the PVP was decomposed at approximately 450 °C (Supplementary Fig. 3), while the metal ions were eventually oxidized by oxygen under ambient air during the calcination step; Ge and Zn ions were thus spontaneously mixed through intermolecular bonds in an amorphous phase (as shown in the TEM image, selected area electron diffraction (SAED) patterns in Fig. 1c and XRD patterns in Supplementary Fig. 4). The amorphous phase was maintained at 500 °C. However, when the calcination temperature increased to 600 and 650 °C, crystalline structures of GeO$_2$ and Zn$_2$GeO$_4$ were gradually developed (Supplementary Fig. 5).

Afterward, the as-calcined NFs underwent a reduction process under H$_2$ gas via a solid–gas reduction reaction at 600 °C to remove oxygen atoms from the NFs (denoted as O-dGZNFs) while maintaining a similar crystal structure in nanoscale (Fig. 1d). The O-dGZNFs had a uniform distribution of Ge–Zn atoms, according to high-angle annular dark-field scanning transmission electron microscopy (HAADF-STEM) mapping (Fig. 1e) without significant morphological changes or agglomeration (Supplementary Fig. 6). As a result of this step, germanium–oxygen (Ge–O) bonds were reduced to Ge–Ge bonds with distorted $d$-spacing, whereas oxygen atom-incorporated Zn (Zn–O) remained in the amorphous phase without reduction owing to insufficient activation energy at the given temperature (Supplementary Figs. 4 and 7). Additionally, higher structural disorder was observed in the O-dGZNFs than in the O-dGNFs without Zn (Supplementary Fig. 4b). The main Ge metal peak of microstructure, representing the (111) plane in a crystal region of the O-dGZNFs, was slightly shifted due to the $d$-spacing increase and separated by the development of new kinds of bonds. We believe that this change is attributed to the atomic-level influence of Zn and structural distortion because atomic radius of Zn (142 pm) is theoretically larger than that of Ge atom (125 pm).

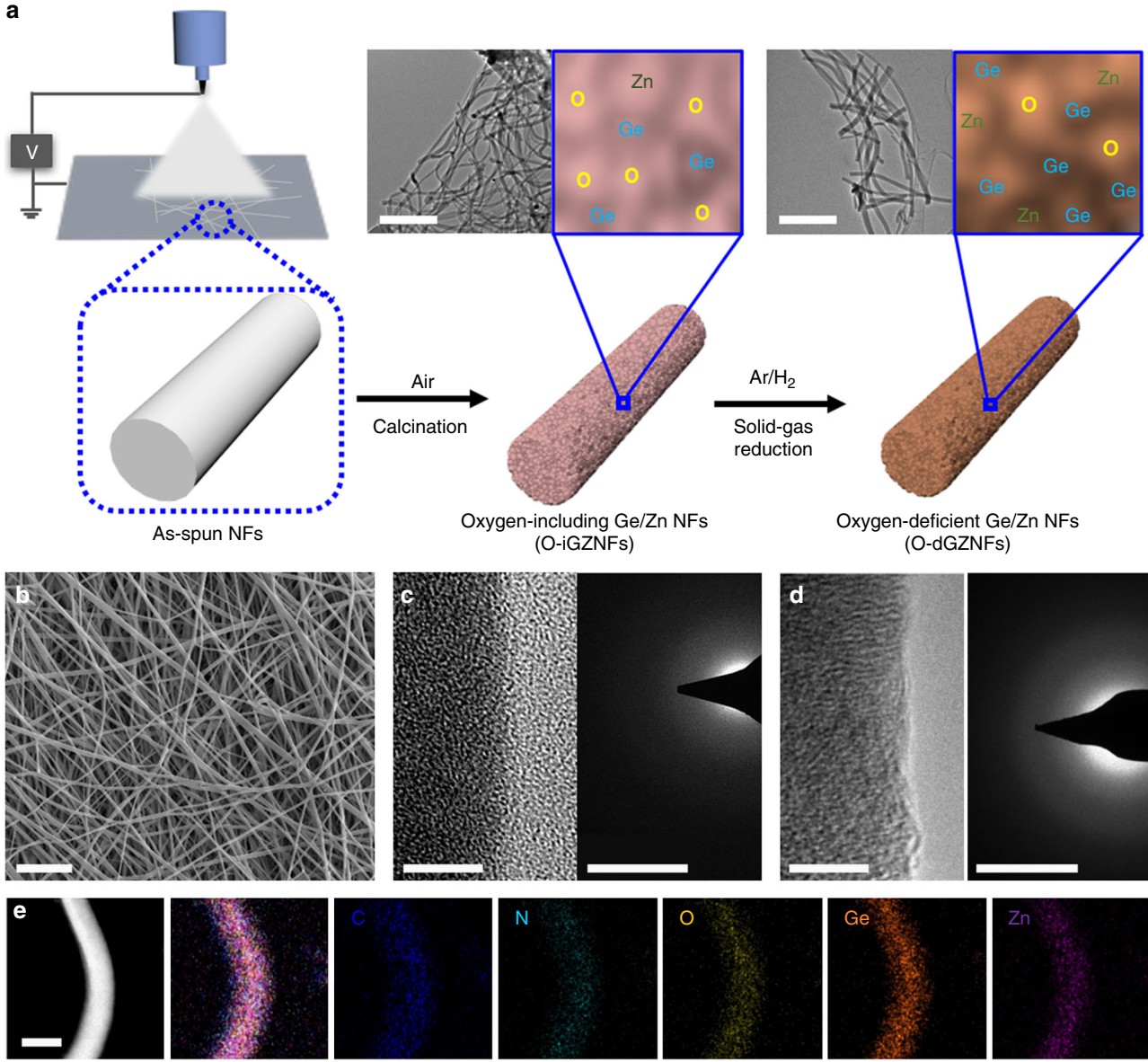

**Fig. 1** Morphological structure evolution. **a** Schematic illustration of the whole synthetic process. The inset TEM images in **a** correspond to O-iGZNFs and O-dGZNFs. **b** SEM image of as-spun NFs. HR-TEM images and SAED patterns of **c** O-iGZNFs and **d** O-dGZNFs. **e** HAADF-STEM mapping of O-dGZNFs: red—carbon, orange—nitrogen, yellow—oxygen, cyan—germanium, and green—zinc. Scale bars: **a** 500 nm, **b** 10 μm, **c**, **d** 5 nm and 5 1/nm, and **e** 50 nm

These XRD patterns directly demonstrate the bonding of Ge with foreign atoms in the NFs. Moreover, the high-resolution TEM (HR-TEM) image in Fig. 1d microscopically shows no clear lattice fringe and amorphous SAED patterns, which attributed to the intermolecular Ge–Zn structure having homogeneous Ge–Zn–O bonding and distorted Ge clusters. The TEM images and SAED patterns of O-iGNFs and O-dGNFs are displayed in Supplementary Fig. 8, showing similar crystal structures and morphology to those of O-iGZNFs and O-dGZNFs.

To accurately confirm the chemical bonding, Raman spectra were obtained as shown in Fig. 2a. The O-dGZNFs showed broad Ge–Ge bonds in the range of 280–305 cm$^{-1}$ and independently oxygen-defective and asymmetric Ge–Zn–O bonds at 750 and 777 cm$^{-1}$, respectively, as well as partial Zn–O bonds (437 cm$^{-1}$). In contrast, Ge–O bonds at 445 cm$^{-1}$ and broad Ge–Ge peaks were detected in the O-dGNFs[30–33]. The Raman spectra of both NFs confirmed that the amorphous-phase carbon was formed by the thermal decomposition of PVP, as determined from the

intensity of the $I_D/I_G$ ratios: 1.46 (O-dGNFs) and 1.30 (O-dGZNFs). These results indicate that oxygen-shared Ge and Zn exist in a state of intermolecular connection in the amorphous carbon matrix. Furthermore, the triggering of Ge–Ge distortion by this featured bond was proven through core-level X-ray photoelectron spectroscopy (XPS) spectra. Fig. 2b, c displays the characteristic peaks related to Ge–Zn–O at approximately 1021, 1044 (Zn 2p), and 32 eV (Ge 3d), different from the case of O-iGNFs and O-dGNFs (Supplementary Fig. 9)[34,35]. Moreover, Fourier transform-extended X-ray absorption fine structure (FT-EXAFS) analyses in Fig. 2d, e reveal that the Ge–Ge bonding length was elongated in the presence of Zn (Zn–Zn: 2.29 Å)[36]. The O-dGNFs showed an increase in the peak intensity of Ge–Ge bonds at 2.14 Å by the loss of Ge–O bonds during the reduction process, which is consistent with the value for reference Ge (2.14 Å)[37,38]. However, the O-dGNFs have still partial GeO$_x$ contents, as confirmed by the XPS results (Supplementary Fig. 9). Meanwhile, the peak positions in the O-dGZNFs were slightly shifted

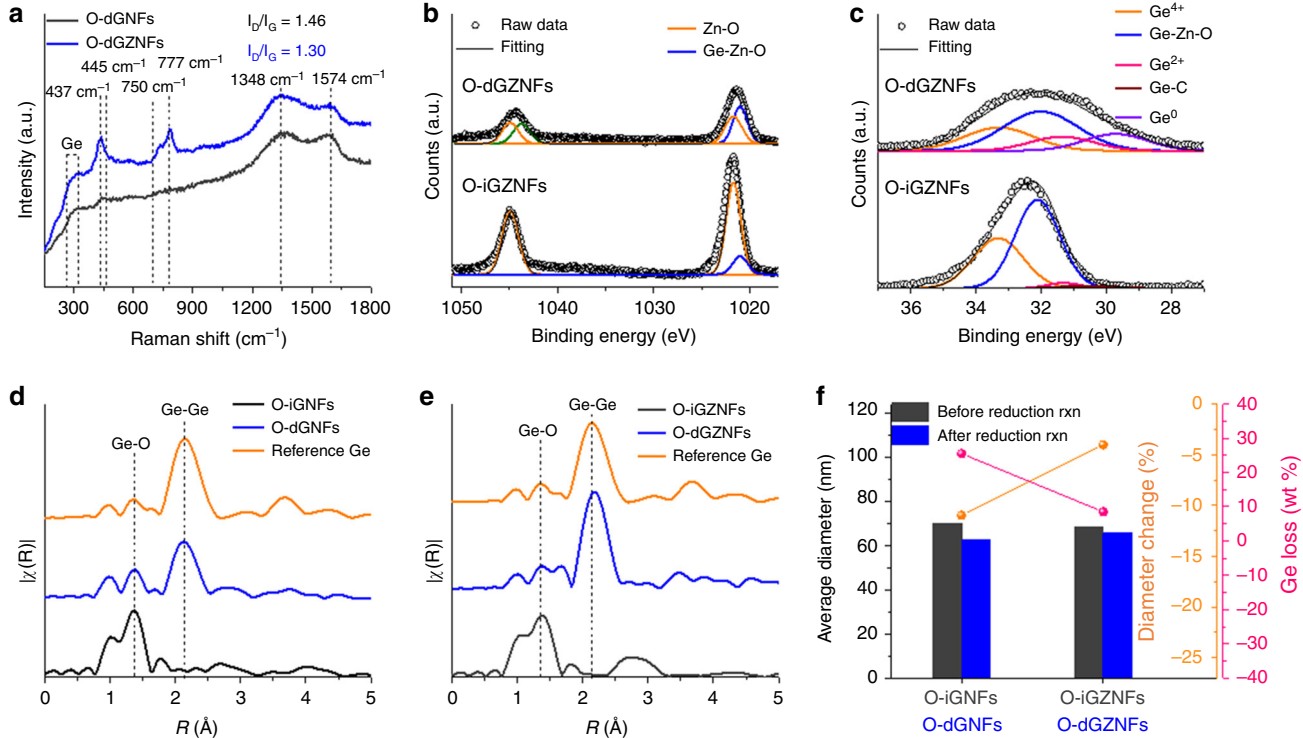

**Fig. 2** Structural analysis of the synthesis. **a** Raman spectra of O-dGNFs and O-dGZNFs. Core-level XPS spectra of O-iGZNFs and O-dGZNFs in **b** Zn 2p and **c** Ge 3d. EXAFS spectra of the **d** O-dGNF and **e** O-dGZNF series. Pure Ge was used as a reference sample. **f** Diameter change and Ge loss in O-dGNFs and O-dGZNFs

to the right (to 1.67 and 2.17 Å from 1.38 and 2.14 Å, respectively) and did not perfectly overlap with Ge–O bonds as well as with the peak related to Ge–Ge bonds. This unusual result is attributed to the disordered Ge–Ge $d$-spacing by Ge–Zn intermolecular connections and the longer bonding length of intrinsic Zn–Zn than that of Ge–Ge. In addition, this delicate interconnection can prevent the chronic problem of Ge sublimation during the reduction of GeO$_2$. GeO$_x$ and/or Ge gas molecules at high temperature can be vaporized without a phase transition to solid-state Ge metal due to their unstable state[39]. Core-level XPS spectra of C, N, and O for the O-iGNFs/O-iGZNFs and O-dGNFs/O-dGZNFs were also analyzed (Supplementary Figs. 10 and 11); C–C bonding became more intense after the reduction process, and Ge–Zn–O bonding was also visible. The calibrated graph in Fig. 2f based on energy-dispersive X-ray (EDX) spectroscopy and SEM images confirmed that the Ge content dramatically decreased during the reduction process owing to the sublimation of unstable Ge states. For this reason, 26 wt% of Ge was lost and 11% of diameter shrinkage occurred, as determined through the comparison of O-iGNFs and O-dGNFs. In contrast, the O-dGZNFs showed distinct characteristics with only 8 wt% of Ge loss and a 4% of diameter change during reduction. The overall weight change (Supplementary Fig. 12) also showed similar patterns between O-dGNFs and O-dGZNFs: the introduction of Zn resulted in a higher yield of Ge. The well-distributed Ge–Zn atoms with Ge–Zn–O bonds resulting from intermolecular interactions clearly bind near Ge–Ge bonds because Zn–O consumes minimal oxygen at this step due to insufficient activation energy for the reaction (ZnO (s) + H$_2$ (g) → Zn (s) + H$_2$O (g)). That is, strong Zn–O bonds directly connected to Ge–O can hold metallic Ge without vaporization. Such intermolecular interactions in O-dGZNFs enhance the electrical properties due to the uniform metallic zinc distribution, leading to high electronic conductivity and ultralong cycle

stability as well as structural maintenance. In addition, the introduction of Zn into O-dGZNFs resulted in a higher surface area (61.3 m$^2$ g$^{-1}$) than that (37.1 m$^2$ g$^{-1}$) of O-dGNFs (Supplementary Fig. 13), with higher amounts of distinct mesopores (between 3 and 4 nm).

**Revealing the electrochemical maturation of O-dGZNFs.** Ge anodes feature intrinsically inferior electronic conductivity and undergo a large volume change during cycling in LIBs, triggering poor electrochemical performance despite exhibiting a high theoretical capacity. In contrast, Zn demonstrates a low volume expansion (Li$_x$Zn, $0 < x < 1$) and outstanding electronic conductivity. Therefore, the O-dGZNFs, a combination of Ge and Zn, could show compatible advantages such as improved electronic conductivity and structural integrity. Galvanostatic measurements of O-dGNF and O-dGZNF electrodes were conducted to determine the dependence of their electrochemical properties on the intermolecular bonding of Ge–Zn. The differential capacity results in Fig. 3a, b show redox peaks during the discharge/charge of O-dGNFs and O-dGZNFs, respectively. In the 1st cycle, an SEI layer was formed, and the conversion reaction of Ge–O occurred at 0.87 V in the cathodic part. Thus, the peaks at approximately 0.18, 0.36, and 0.51 V are related to the Li–Ge alloy reaction. Furthermore, Li ions were extracted from the Li–Ge alloy at the anodic peaks (0.36 and 0.49 V), followed by the reformation of Ge–O bonds due to lithium oxide (Li$_2$O) decomposition at 1.08 V. The O-dGZNF electrode displayed a similar tendency, but some peaks were slightly shifted during the 1st cycle, which can be ascribed to the introduction of foreign Zn atom. After the 1st cycle, the cathodic and anodic peaks of both electrodes remained consistent. Fig. 3c exhibits the first galvanostatic discharge/charge capacities of O-dGNFs (1965/1360 mAh g$^{-1}$) and O-dGZNFs (1444/1043 mAh g$^{-1}$), which show a low initial Coulombic efficiency (ICE) of 69.2% and 72.2%,

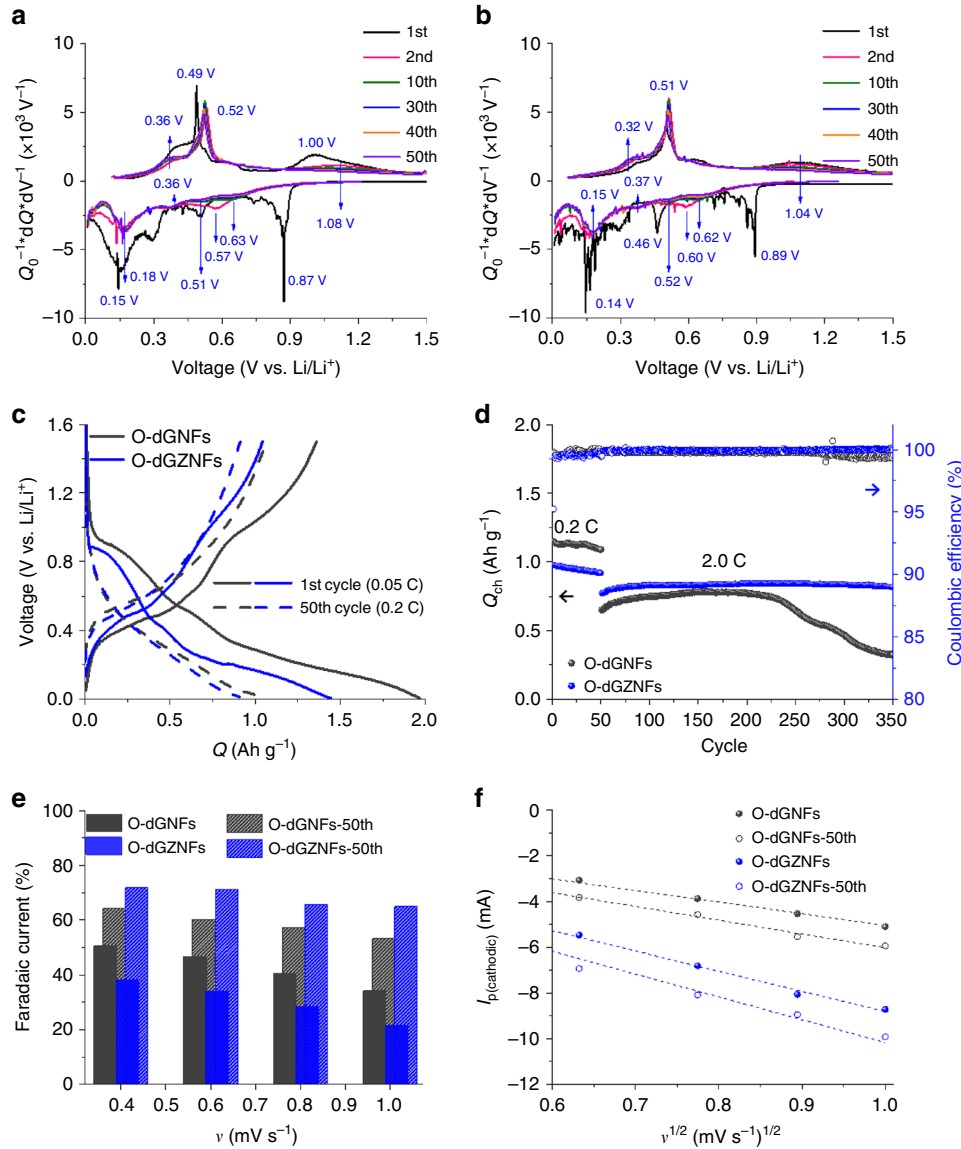

**Fig. 3** Electrochemical properties of O-dGNF and O-dGZNF electrodes. Differential capacities of **a** O-dGNFs and **b** O-dGZNFs for featured cycles. **c** Discharge/charge profiles at the 1st cycle (bold line) and 50th cycle (dashed line). **d** Charging capacity retention at 0.2 and 2.0 C-rate. **e** Comparison of the faradaic current of pristine and 50-cycled electrodes at various scan rates. **f** Plots of squared scan rate vs. peak current calculated by the Randles–Sevcik equation (Eq. (2))

respectively, because one-dimensional materials feature a large surface area, which means that more Li ions participate in the formation of SEI layers. The introduction of Zn resulted in a slightly enhanced ICE, although the difference was not significant. Surprisingly, the galvanostatic discharge/charge profile of the 50th cycle did not show a plateau related to conversion and $Li_2O$ decomposition compared with that of the 1st cycle. Moreover, the $Li_2O$ decomposition at 1.08 V (O-dGNFs) and 1.04 V (O-dGZNFs) decreased and vanished after cycles (Fig. 3a, b). This trend suggests that the oxygen components in the NFs were transformed into different forms during the initial cycles. Core-level XPS spectra of 50-cycled electrodes (O-dGNFs-50th and O-dGZNFs-50th) were characterized after the removal of partial SEI layers by slight etching. As shown in Supplementary Fig. 14a–c, most of the oxygen components in the NFs were reformed into $Li_2O$ and lithium carbonate ($Li_2CO_3$) composites, as well as components of SEI layers[40–42]. Additionally, the presence of LiF in nanofibers is attributed to the decomposition of $LiPF_6$ and reduction of fluoroethylene carbonate (FEC) due to the lower

unoccupied molecular orbital (LUMO) level of LiF, which is in accordance with previous works (Supplementary Fig. 14c, d)[43–46]. Furthermore, Ge–Zn–O and Ge metallic bonds remained after the 50th cycle. The capacity retention (Q) with cycling in Fig. 3d reflects oxygen deformation and irreversible $Li_2O$ decomposition during the initial 50 cycles as the capacity gradually decreased.

The above phenomenon influences the electrochemical properties of NFs. After the 50th cycle, the O-dGZNFs displayed outstanding reversible capacity and cycle retention until the 350th cycles, even at 2.0 C-rate, with almost 100% capacity retention. In addition, to examine the effect of oxygen content on the electrochemical performance of the O-dGZNF electrode as well as to determine the optimized oxygen contents of the NFs, charge capacity retention tests (Supplementary Fig. 15) were carried out for O-dGZNFs with different Ge/Zn ratios; a higher proportion of Zn led to a higher oxygen content, resulting in stable cycle retention owing to the high ionic/electronic conductivity but poor reversible capacity; a lower oxygen content resulted in a higher initial capacity, but capacity fading was more prominent. In

contrast, the O-dGNFs required an activation step to attain optimal capacity, showing a capacity gain during tens of cycles. However, the O-dGNF electrode subsequently suffered from fatal capacity decay with only 49.5% retention due to the low electronic conductivity and low response to current changes. To verify this behavior and confirm the main reasons, cyclic voltammetry (CV) measurements were conducted at various scan rates after the 1st cycle and 50th cycle. The results are shown in Supplementary Fig. 16; the contributions of faradaic and capacitive (nonfaradaic) current were calculated using Eq. (1) and are shown in Fig. 3e:

$$i_p = k_1 \nu + k_2 \nu^{1/2} \tag{1}$$

where $i_p$ is the peak current and $\nu$ is the scan rate in the CV graph, and $k_1$ and $k_2$ are the constants for the capacitive and faradaic current, respectively. The capacitive current is related to surface reactions on the electrode, while the faradaic current is related to the charge transfer redox reaction at the electrode. In other words, the total current in CV measurements consists of faradaic and capacitive currents. Fig. 3e shows that the capacitive current was dominant for both electrodes at all scan rates from 0.4 to 1.0 mV s$^{-1}$ due to the high surface area of the one-dimensional structure (capacitive current: 65.8% for O-dGNFs and 78.5% for O-dGZNFs). In particular, the O-dGZNF electrode, which has a higher BET surface area, showed a higher capacitive current than that of the O-dGNF electrode. However, the dominant current switched to the faradaic current after the 50th cycle for both electrodes; this change can be ascribed to the formation of lithium composites, such as $Li_2O$ and $Li_2CO_3$, as well as metallic bonds (Ge–Ge or Ge–Zn). In particular, Zn has better intrinsic electronic conductivity than Ge, leading to a higher charge transfer rate and a greater faradaic current in O-dGZNFs. In addition, the lithium composites detected in the XPS results in Supplementary Fig. 14a–c act as ion conductors that can enhance the ion diffusion rate in the electrode[44–46]. For an evidence, we calculated the Li ion diffusion coefficient by linearly fitting the plots of the square root of the scan rate ($\nu^{1/2}$) with respect to the peak current ($I_p$) (Fig. 3f) based on the Randles–Sevcik equation (Eq. (2)):

$$i_p = 268,600 n^{3/2} A D^{1/2} C \nu^{1/2} \tag{2}$$

where $n$ is the number of transported electrons, $A$ is the area of the electrodes in a coin-type cell, $D$ is the diffusion coefficient of Li ions in the electrode, and $C$ is the concentration of Li salt in the electrolyte. The slope, which is related to the Li ion diffusion coefficient, increased after 50 cycles as the ion diffusivity increased (Fig. 3f). The Li ion diffusion coefficients of cycled electrodes were 1.2–1.4 times higher than those of pristine electrodes. Such increased Li ion diffusion coefficients suggest better ionic conductivity arising from the formation of lithium composites ($Li_2O$ and $Li_2CO_3$) during cycling. After the 1st, 10th, and 50th cycles, the Li ion diffusion coefficients gradually increased; this increase was proportional to the increased amount of lithium composites (Supplementary Fig. 17 and Supplementary Table 1).

Regarding the relationship between metallic bonds and electronic conductivity, the existing Ge–O–Ge or Ge–Zn–O bonds would be reduced to Ge–Ge or Ge–Zn when the oxygen atoms are combined with lithium and carbon during cycling. The XPS spectra in Fig. 4a, b and Supplementary Fig. 14e show the development of more Ge$^0$ bonds and peak shifts compared to those observed in the XPS results of O-dGNFs and O-dGZNFs in Fig. 2b, c and Supplementary Fig. 9. In particular, the Ge–Zn–O peak in O-dGZNFs was located at approximately 32 eV (Ge 3d) and 1021/1044 eV (Zn 2p), but all peaks of Ge–Zn–O were

partially shifted, to 31.5, 1020.6, and 1043.4 eV, arising from oxygen defects within the bonds. The gradual appearance of metallic bonds in the NFs, corroborated by XPS analysis, can also account for the increased faradaic current because electronic conductivity is also enhanced with increased metallic bonds and Li ion diffusion. Furthermore, FT-EXAFS analysis proved the maintenance of intermolecular bonding and oxygen removal in O-dGNFs-50th and O-dGZNFs-50th in various charge/discharge states compared with the corresponding pristine states before cycling (Fig. 4c, d). In accordance with the trends mentioned above, the intensity of Ge–O significantly decreased for both O-dGNFs-50th and O-dGZNFs-50th. The peak of O-dGNFs-50th coincides with the reference Ge peak (2.14 Å), whereas O-dGZNFs-50th still exhibits a peak at 2.17 Å; these results demonstrate the reversible characteristics of Ge–Zn bonds during cycling. When both electrodes were discharged, the Ge–Ge peak was relocated to 2.19 and 2.20 Å, signifying the emergence of Li–Ge bonds from metallic Ge–Ge and Ge–Zn bonds; this result is also supported by X-ray absorption near-edge structure (XANES) profiles (Supplementary Fig. 18)[39,47]. To further compare the electrochemical/structural stability of O-dGZNFs with that of O-dGNFs, real-time characterizations were conducted using in situ TEM and EIS.

**In situ observation of the structural and electrochemical behavior of O-dGZNFs.** To demonstrate the behavior of a single NF during cycling in real time, in situ TEM experiment was carried out with *I–V* curve analysis to examine the structural and phase evolution of an O-dGZNF as well as the electronic conductivity of a single NF during the electrochemical discharge/charge process[48]. The fine structure of the Ge–Zn distorted array can unpredictably affect the structural stability and electrochemical properties throughout the whole process. For in situ TEM, all potentiostatic measurements were carried out using a nanobattery that consisted of a single NF-based working electrode (sample) and counter electrode (Li metal) with naturally formed $Li_2O$ as a solid-state electrolyte on the Li metal, as illustrated in Supplementary Fig. 19. Time-resolved snapshots of the O-dGZNF were taken at 0, 900, 982, 1144, and 1897 s, showing the morphological evolution of the material from the discharged state to the charged state (Fig. 5a–e). Before lithiation, the O-dGZNF was 63 nm in diameter (Fig. 5a, k) with no noticeable Ge diffraction patterns due to the distorted structure and amorphous rings (Fig. 5f). When a negative bias of −2 V was applied to initiate lithiation (discharge) process on the O-dGZNF, $Li_2O$ was initially formed due to the native oxide layer and oxygen atoms that came from Ge–Zn–O bonds (Fig. 5b, g). Then, upon further lithiation, Ge and Zn atoms solely reacted with Li ions by forming Li$_x$Ge and LiZn bonds, as shown in Fig. 5c, h[43]. Finally, crystalline $Li_{15}Ge_4$ was completely formed from amorphous Li$_x$Ge and showed a diameter of 91 nm, indicating 44% expansion (Fig. 5d, i, l and Supplementary Movie 1)[49]. After 1144 s, delithiation (charge process) was launched by applying a positive bias of 2 V, and Li was extracted from the O-dGZNF (Supplementary Movie 2). Afterward, the diameter of the NF was reduced to 82 nm (31% expansion with respect to pristine NF), while considerable amounts of Li–Ge and Li–Zn alloys were converted back to amorphous Ge (a-Ge) and Ge–Zn (Fig. 5e, j, m); these results corresponded to the XPS results of cycled samples in Fig. 4 and Supplementary Fig. 14. The overall volume change of the NF during one cycle in real time is shown in Fig. 5n. For comparison, in situ TEM was also conducted for the electrochemical discharge process of an O-dGNF under the same conditions. Although the phase transition (formation of $Li_2O$ and alloyed Ge) was similar upon lithiation, the overall volume expansion after delithiation

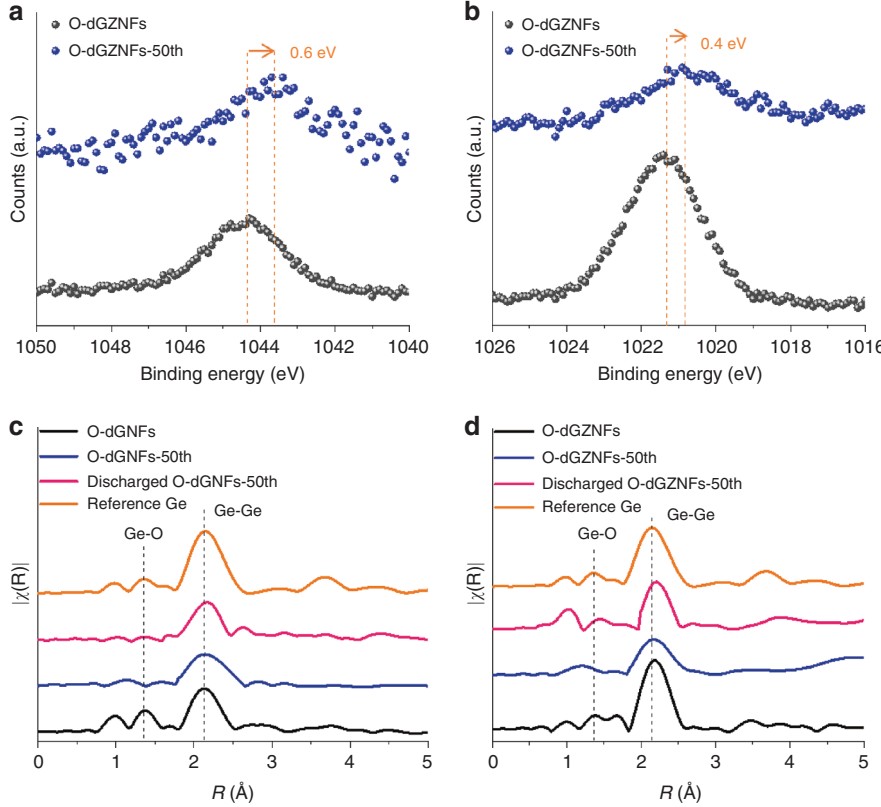

**Fig. 4** Physical analysis of O-dGNFs and O-dGZNFs at various states. **a**, **b** Core-level XPS spectra of pristine and O-dGZNFs-50th electrodes after partial etching to remove the SEI layer in Zn 2p. EXAFS spectra of **c** O-dGNFs and **d** O-dGZNFs at various states

was larger (51% diameter change), even though the NF was thinner (Supplementary Fig. 20 and Supplementary Movie 3). Based on these results, Zn helped to alleviate volume expansion; this assistance can be ascribed to better structural integrity arising from the formation of Ge–Zn–O and a lower theoretical volume change in Zn.

To comprehend such trends at the bulk scale, ex situ SEM analysis was performed to further compare the degrees of volume change in O-dGZNFs and O-dGNFs. The cross-sectional SEM images in Supplementary Fig. 21 showed a smaller thickness change in O-dGZNFs (11.6%) than in O-dGNFs (13.3%) after the 1st cycle. The intermolecular introduction of Zn atoms, which partially act as buffering layers, into the Ge-based NFs resulted in a lower degree of volume change. As a result, the O-dGNF electrode finally underwent a 211% volume expansion after 300 cycles due to the agglomeration of Ge particles and fracturing during cycling, whereas the O-dGZNF electrode only showed a 100% volume change. In addition, ex situ TEM results after 50 cycles revealed that more stable SEI layer on the surface of O-dGZNFs was formed compared with that of the O-dGNFs due to the introduction of Zn with highly stable interfacial properties (Supplementary Fig. 22). The large volume expansion of O-dGNFs was attributed to the pulverization of the active materials, leading to unsustainable cycle retention (Fig. 3d). In addition, the electronic characteristics of O-dGNF and O-dGZNF were verified through in situ conductivity measurements (Fig. 5o) of each single NF. The conductivity was measured at three different states (state i, ii, and iii, corresponding to Fig. 5f, g, i, respectively) of lithiation in a voltage range from −10 to 10 V. When we simply calculated the electrical conductance using the Ohm's law ($R = V/I$ or $G = I/V$, where $R$, $G$, $V$, and $I$ indicate the electrical resistance, conductance, voltage, and current, respectively), a small difference in conductance between the two single NFs was

apparent in state i. In contrast, the electrical conductance of the O-dGZNF in state ii was 36 times higher than that of O-dGNF because metallic Ge–Zn bonds were formed due to $Li_2O$ formation from Ge–Zn–O. Furthermore, the conductance of O-dGZNF in state iii was still 31 times higher than that of O-dGNF even after complete lithiation of both NFs—this result is attributed to the higher intrinsic conductivity of Zn ($1.7 \times 10^7$ S m$^{-1}$) than that of Ge ($2 \times 10^3$ S m$^{-1}$) in the bulk; furthermore, lithiated Ge exhibits enhanced electrical properties. This result clearly demonstrates that the intermolecular introduction of Zn leads to dramatically improved electrical conductance and can be expected to provide a facile electron pathway upon discharge/charge.

Additionally, in situ galvanostatic electrochemical impedance spectroscopy (GS-EIS) (Fig. 5p) was performed at 1.0 C-rate during lithiation to obtain a comprehensive understanding of the effect of Zn on electrochemical performance because the electrochemical kinetics of electrodes are strongly related to the charge transfer resistance ($R_{CT}$). The overall $R_{CT}$ value of the O-dGZNF electrode including intermolecular Zn was significantly less than that of the O-dGNF electrode owing to the enhanced electronic conductivity of O-dGZNFs. Moreover, the improved ionic conductivity resulting from the random distribution of $Li_2O$ and $Li_2CO_3$ in NFs helps Li ions easily penetrate the active materials and maintain a low $R_{CT}$ value. However, $R_{CT}$ typically increases in the stage of deep lithiation due to the formation of insulating $Li_2O$ and $Li_2CO_3$, which interrupt the electron transport, as shown in O-dGNF electrode. In contrast, the O-dGZNF electrode sustained a low value of $R_{CT}$ regardless of the formation of $Li_2O/Li_2CO_3$, attributed to the intermolecular distribution of Zn which exhibits outstanding electronic conductivity. These results suggest that randomly formed lithium composites decrease the resistance with an elevated ion diffusion rate and that the intermolecular interactions of Zn in the

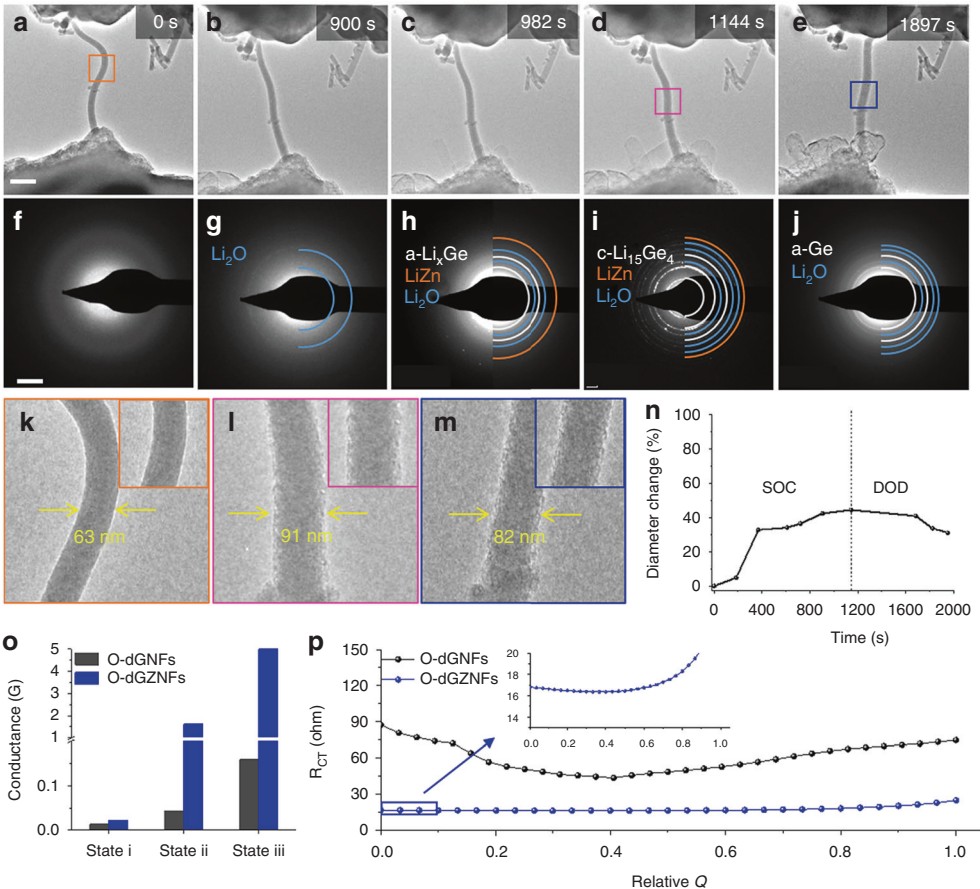

**Fig. 5** In situ characterization of O-dGNFs and O-dGZNFs. In situ TEM observations were conducted with an O-dGZNF sample. Time-resolved TEM images for **a–d** lithium insertion and **e** lithium extraction in real time. **f–j** Each SAED pattern corresponds to the TEM image above. **k–m** Magnified TEM images of featured states. **n** Curve of diameter change vs. time upon lithiation/delithiation. **o** In situ electrical conductivity measurement of O-dGNFs and O-dGZNFs during the lithiation process at three points, marked as state i, ii, and iii and corresponding to **f**, **g**, and **i**, respectively. **p** In situ EIS galvanostatic measurements during lithiation at 1.0 C-rate of O-dGNFs and O-dGZNFs. Scale bars: **a** 200 nm and **f** 5 1/nm

electrode significantly provide a facile electron pathway while simultaneously suppressing a resistance increase.

**Extensive electrochemical performance characterization and practical confirmation of O-dGZNFs.** The unstable structure of Ge suffers from electrochemical volume expansion during lithiation, which triggers undesired side reactions, such as delamination and pulverization of the electrode with the continuous consumption of electrolyte. The structural evolution of intermolecularly connected O-dGZNFs exhibited outstanding electrochemical performance under harsh conditions (high C-rates) and ultralong cycle retention at 3.0 C-rate. The morphological structure of O-dGNFs was broken down due to uncontrollable volume expansion with uneven interfacial properties (Supplementary Fig. 22a), which led to poor rate capability (Fig. 6a). Such unstable structures inhibit facile electron transport at high current rates, leading to rapid capacity fading at high C-rates. In contrast, the O-dGZNFs became porous and hollow over the course of 50 cycles while relatively uniform interfacial layers containing $Li_2O$ and $Li_2CO_3$ were formed, and the structural integrity of the material was maintained (Supplementary Fig. 22b). As a result, O-dGZNFs can provide fast electron/ionic transport at high current rates and retain considerable capacity (~50% at 20 C compared with that at 0.2 C-rate). Furthermore, the O-dGZNFs showed a capacity recovery of 98.7% when the C-rate changed from 20 C to 0.2 C-rate. As O-dGZNFs possess suitable electrode architectures that allow facile ionic/electronic transport with

considerable structural stability, this material also exhibits ultralong cyclability (capacity retention of 73%) at high C-rates (3.0 C-rate) without any dramatic capacity decay from electrode delamination or pulverization (Fig. 6b).

Due to their practical feasibility, both O-dGNFs and O-dGZNFs were finally paired with a $LiCoO_2$ (LCO) cathode, which exhibited an electrode thickness of approximately 52.2 μm and typical electrochemical characteristics (Supplementary Fig. 23). Because one-dimensional materials generally form a considerable proportion of the SEI layer due to their high surface area, the anodes were treated through a prelithiation process before full cell assembly. The full cells of O-dGNFs and O-dGZNFs initially exhibited reversible capacities of 2.11 and 2.02 mAh cm$^{-2}$ after the formation cycle (Supplementary Fig. 24a). Discharge capacity of both electrodes is higher than initial charge capacity owing to prelithiated anodes. Further cycle retention tests at 1.0 C-rate (Fig. 6c) confirmed the significantly superior performance of O-dGZNFs when assembled in a full cell, delivering a reversible areal capacity of 0.95 mAh cm$^{-2}$ with a capacity retention of 60% for 400 cycles, in contrast to O-dGNFs (capacity retention of only 23% with an areal capacity of <0.5 mAh cm$^{-2}$). The demonstration of LED light is further showcased for O-dGZNFs in Supplementary Movie 4, which highlights the practical operation of such full cells. Moreover, the rate capabilities of O-dGZNFs and O-dGNFs assembled in full cells were further characterized and are shown in Supplementary Fig. 24b; superior rate

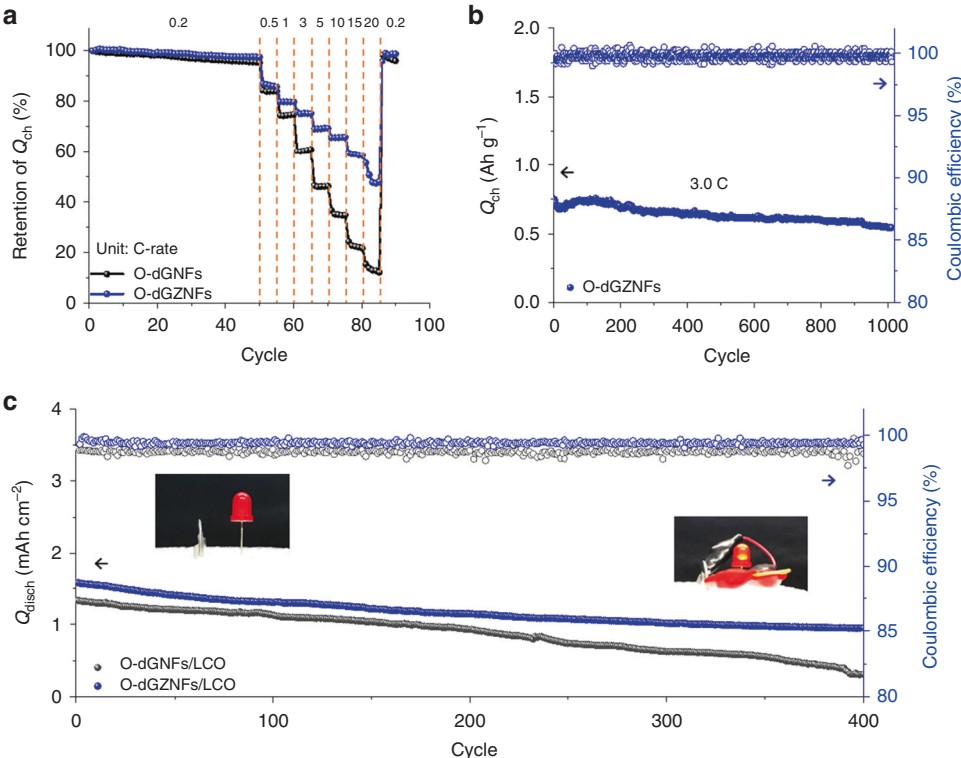

**Fig. 6** Rate capability and long-term cyclic stability in both half and full cells. **a** Rate capability of O-dGNFs and O-dGZNFs. **b** Cyclic performance of O-dGZNFs at 3.0 C-rate. **c** Electrochemical performance of full cells assembled with LCO at 1.0 C-rate. Inset photographs indicate the charged O-dGZNFs/LCO full cell before and after closing the circuit

capabilities were also observed for O-dGZNFs in assembled full cells.

## Discussion

Ge and Zn atoms are homogeneously distributed in our one-dimensional, oxygen-deficient, disordered Ge–Zn NFs (O-dGZNFs). The delicate formation of Ge–Zn bonds suppressed Ge sublimation during the synthetic process, which resulted in a higher yield of Ge after the reduction process. The intermolecular interactions of Ge–Zn in NFs not only contributed to enhanced structural integrity but also enabled faster electron and ionic transport, as evidenced by the higher electronic conductivity and Li ion diffusivity of O-dGZNFs compared with those of O-dGNFs. In addition to the superior architectures of O-dGZNFs compared with those of O-dGNFs, more uniform interfacial layers were formed on the O-dGZNFs, which additionally act as fast ionic conductors. As the dual functionalities of Zn not only improve electronic conductivity but also maintain structural integrity, the O-dGZNFs exhibited outstanding rate capability (~50% capacity retention between 0.2 and 20 C-rate) and cycle retention (73% at 3.0 C-rate) with 546 mAh g$^{-1}$ even after 1000 cycles. Furthermore, stable full cell operation of O-dGZNFs was demonstrated with LCO as a cathode for 400 cycles at a relatively high C-rate (1.0 C-rate). This work sheds light on the utilization of intermolecular interactions of metal atoms (Ge and Zn) to ensure facile ionic/electronic transport and stable interfaces/morphologies. This approach could be extended to other electrode materials to enable advanced rechargeable energy storage systems.

## Methods

**Materials**. Germanium oxide (GeO$_2$, 99.995%) was purchased from Kojundo Chemical Lab. Co., Ltd. Poly(vinyl pyrrolidone) (PVP, $M_w \sim 1,300,000$ g mol$^{-1}$) and zinc nitrate hexahydrate (Zn(NO$_3$)$_2$·6H$_2$O, reagent, 98%) were purchased from Sigma Aldrich. Germanium (Ge) was purchased from Alfa Aesar for FT-EXAFS

measurement of reference Ge. All the chemicals were used without further purification.

**Synthesis of O-iGNFs and O-iGZNFs**. O-iGNFs and O-iGZNFs were prepared via electrospinning and a subsequent calcination step. For the preparation of O-iGNFs, 0.4 g of GeO$_2$ was dissolved in 50 mL of deionized (DI) water at 90 °C. Then, 7.5 g of PVP was added into the solution and stirred at 500 rpm for 6 h. Then, the electrospinning solution was loaded into a syringe, where the electrospinning process took place using an electrospinning tool (Machine 1 Type, NanoNC). The electrospinning was conducted with the following conditions: a flow rate of 0.5 mL h$^{-1}$, an applied voltage of 16.0 kV, and a distance of 15 cm between the tip of the syringe and the current collector using a 25-gauge needle. The as-spun NFs were calcined at 250 °C for 1 h and 500 °C for 2 h at a ramping rate of 5 °C min$^{-1}$ in a box furnace (MF-22G, JEIO TECH). For the preparation of O-iGZNFs, a solution mixture with specific amounts of GeO$_2$ and Zn (NO$_3$)$_2$·6H$_2$O (98%) (Supplementary Fig. 2b) was added to the electrospinning solution and subjected to identical electrospinning conditions and subsequent calcination. As a result, O-iGNFs and O-iGZNFs were obtained.

**Synthesis of O-dGNFs and O-dGZNFs**. As-synthesized O-iGNFs and O-iGZNFs underwent a reduction process in a quartz furnace (OTF-1200X-II, MTI Corporation) filled with argon (Ar). In this process, the furnace was heated to 600 °C under Ar at a ramping rate of 5 °C min$^{-1}$, and once the temperature reached an expected value, the atmosphere was changed to an Ar and hydrogen gas mixture (Ar/H$_2$ (96/4, v/v)) and maintained for 1 h. Afterward, the furnace was filled again with Ar instead of Ar/H$_2$ and cooled spontaneously to room temperature.

**Materials characterization**. Morphological images were acquired by field-emission scanning electron microscopy (FE-SEM, Nova 230, FEI) operating at 10 kV and environmental transmission electron microscopy (ETEM, FEI) with an accelerating voltage of 300 kV under vacuum conditions. In addition, energy-dispersive X-ray (EDX) detection combined with scanning transmission electron microscopy (STEM, FEI) was used to analyze the elemental distribution and the amounts of elements. Raman spectra were collected with a confocal Raman spectrometer (alpha 300R, WITec) with a laser wavelength of 532 nm. XRD patterns were obtained by using a Bruker D8-advance with 3 kW Cu Kα radiation and wavelengths in the range of 20–80°. An inductively coupled plasma (ICP) instrument combined with a mass spectrometer (LC-ICP/MS, PerkinElmer, ELAN DRC-II) and elemental analyzer (EA, Flash 2000) were used to examine the elemental content. X-ray photoelectron spectroscopy (XPS) (K-alpha, ThermoFisher) was

used for surface analysis. X-ray absorption near-edge structure (XANES) and extended X-ray absorption fine structure (EXAFS) analyses were conducted at the BL6D beamline of the Pohang light source (PLS-II) under a current of 300 mA with 3.0 GeV.

**In situ electrochemical observations**. For in situ TEM measurements using ETEM, an open cell, which was composed of Li metal for the counter electrode and samples for the working electrode, was manufactured for the observation of electrochemical reactions in real time. This open cell could be operated with a scanning tunneling microscope (STM) holder. The working electrode was held on an platinum (Pt) nanowire, and the counter electrode was loaded on a W tip in an Ar-filled glove box on each side of the holder. Then, the holder was transferred to the ETEM while exposed to air for less than 2 s. During the movement, a very thin $Li_2O$ layer was coated on the Li metal and used as the solid electrolyte. For $Li^+$ ion insertion/extraction, a negative/positive potential was applied to the STM holder by an external bias, $-2$ V/2 V for Li ion insertion/extraction. A negative bias drove $Li^+$ to the samples across the $Li_2O$ layer on the Li metal via the potential difference between the two electrodes, and the samples reacted with $Li^+$, consistent with lithiation. In contrast, a positive bias induced the delithiation step by extracting $Li^+$ from the samples. The in situ conductivity test was almost the same as the in situ lithiation/delithiation observation, except there was no loading of Li metal. Individual single NFs loaded on Pt nanowires could be directly contacted with only the tungsten (W) probe, and an external bias was applied in the range of $-10$ V to 10 V to measure the electrical conductivity. For in situ EIS, a set of multiple impedance spectra was measured by galvanostatic EIS in constant current mode (1.0 C) during lithiation/delithiation since the potentiostat of the in situ EIS system consisted of two different channels: one channel was for measuring impedance spectra, and the other channel was for recording voltage profiles. Input signals were generated by the superposition of sinusoidal current waves of 10 mA amplitude at 200 kHz to 1 Hz (VSP-300, BioLogic).

**Electrochemical measurements**. To prepare the electrodes, a viscous slurry consisting of the active materials, super-P carbon black as a conductive material, and poly(acrylic acid) (PAA)/carboxymethyl cellulose (CMC) (1:1 weight ratio) as a binder with a weight ratio of 70:15:15 (w/w/w) was cast on Cu foil with a loading mass of 1.0–1.3 mg cm$^{-2}$. Then, the electrode was transferred into an Ar-filled glove box to act as the working electrode in 2016-type coin cells (Welcos) with Li metal as a counter electrode, a polypropylene membrane (Celgard) as a separator, and ethylene carbonate/diethyl carbonate (EC/DEC = 3:7, v/v) including 10 wt% fluoroethylene carbonate (FEC) and 1.3 M LiPF$_6$ salt as a liquid electrolyte. Galvanostatic measurements of half cells were conducted using a battery cycler (Wanatech, WBCS-3000) in the range of 0.005–1.5 V (1st cycle) and 0.01–1.5 V (further cycles) at 25 °C. For full cells, Li metal was substituted with LiCoO$_2$ (LCO), while all the other components were identical. The LCO electrode was manufactured with LCO:Super-P:polyvinylidene fluoride (PVdF) binder with a weight ratio of 90:5:5 (w/w/w) and cast on Al foil (loading mass: 13 mg cm$^{-2}$). The full cells were evaluated in the operating voltage range from 2.5 to 4.29 V at 25 °C.

## Data availability
The authors declare that the data supporting the findings of this study are available within the article and its Supplementary Information files. All other relevant data supporting the findings of this study are available on request.

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

## Acknowledgements

This work was supported by the Center for Advanced Soft-Electronics funded by the Ministry of Science, ICT and Future Planning as a Global Frontier Project (CASE-2015M3A6A5072945), and the Pohang Accelerator Laboratory provided the synchrotron radiation source at the 6D beamline used for this study. I.-D.K. was supported by National Research Foundation of Korea (NRF), Grant No. 2014R1A4A1003712 (BRL Program), and the Wearable Platform Materials Technology Center (WMC), funded by an NRF Grant by the Korean Government (MSIP) (No. 2016R1A5A1009926). J.Y.C. was supported by an NRF grant funded by the Korean Government (NRF-2017H1A2A1042006-Global Ph. D. Fellowship Program). C.W. thanks the support of the Assistant Secretary for Energy Efficiency and Renewable Energy, Office of Vehicle Technologies of the U.S. Department of Energy (DOE), under Contract No. DE-AC02-05CH11231, Subcontracts No. 18769 and No. 6951379, under the Advanced Battery Materials Research (BMR) program. The microscopic analysis in this work was conducted at the William R. Wiley Environmental Molecular Sciences Laboratory (EMSL), a national scientific user facility sponsored by DOE's Office of Biological and Environmental Research and located at the Pacific Northwest National Laboratory (PNNL). PNNL is operated by Battelle for the DOE under Contract DE-AC05-76RLO1830.

## Author contributions

S.P. conceived the concept. G.S. and J.Y.C. designed and synthesized the materials, and G.S., S.C., J.R., S.K., W.-J.S., and C.K. conducted structural characterization and electrochemical testing. G.S., L.L., and C.W. observed the in situ TEM electrochemical measurements. C.H. and H.-K.S. performed in situ EIS characterization. G.S., J.Y.C., C.W., I-D.K., and S.P. cowrote the manuscript. All authors discussed the results and the manuscript.

## Additional information

**Competing interests:** The authors declare no competing interests.

