## [Peer Review File · Nature Communications]

Reviewers' comments:

Reviewer #1 (Remarks to the Author):

In this work, the authors report atomic-scale combination of germanium-zinc distorted alloy. These one-dimensional Ge-based materials with intermolecular incorporation of Zn into disordered Ge clusters can greatly enhance both structural integrity and electronic conductivity. Further, the unique features are clearly confirmed through in situ analysis. Besides, as-prepared battery anodes remarkably highlights outstanding rate capabilities and cycle retention with a capacity of 546 mAh g⁻¹ even after 1000 cycles. When assembled in a full cell, it notably facilitates considerable energy density during 400 cycles with 99.4% of average coulombic efficiency. We think it's a remarkable work, however there are still some problems must be figured out before it is published in the high-level journal. We suggest it can be published after major revision.

1. The existence of Zn can prevent a fatal sublimation of germanium/germanium oxide during gas-solid phase reduction reaction, however it can't be testified in Supplementary Fig. 1, the Ge content in O-dGNFs is still quite high, please explain. The sintering temperature is below the melting point of Ge, how to explain the sublimation of Ge occurs?
2. Why Zn is selected? If other metallic elements have similar effect?
3. The sintering temperature is higher than 450 °C, why there is still much carbon remained in the materials?
4. The statement and analysis of the bonding between Ge, Zn and O are ambiguous, please improve it. Some statement seems inconsistent with the data, for example the peak at 750 cm⁻¹ is almost sightless in O-dGZNFs, please modify or give some explanation.
5. How the O element content influence the electrochemical performance of O-dGZNFs, is there any experimental data to support it?
6. Why the high temperature sintering treatment doesn't make the Ge and Zn become crystalline state?
7. The annotation in Fig. 4c seems to have some mistake.
8. The Zn also suffers volume expansion during lithiation, why the volume fluctuation of O-dGZNFs is significantly smaller than that of O-dGNFs?
9. Why the conductance of O-dGZNF in state iii was still 31 times higher than that of O-dGNF even after complete lithiation of both NFs? In this state the Zn is transformed to Li-Zn.
10. As the authors state, the Li₂O and Li₂CO₃ have important impact to the ionic conductivity and conductance of the materials, if some quantitative experimental data can be supplied to support the conclusion?

Reviewer #2 (Remarks to the Author):

This work reported the preparing of Ge-Zn nanofiber and studied its lithium storage properties. It seems that the as-prepared battery anodes shows improved lithium storage properties, including rate capabilities and cycle performance. This manuscript is good at materials characterization but the results analysis is not sufficient. I don't recommend it is accepted. There are several important points I wish to bring up.

1. Are there carbon and nitrogen element in Ge-Zn nanofiber? Carbon and nitrogen element can be seen in Fig. 1e. This is very important.
2. The carbon derived from PVP also improve the electronic conductivity.
3. Why the capacitive contribution of o-dGNFs is higher than o-dGZNFs? Why the capacitive contribution was increased after 50th cycles? Please check the particle size before and after 50th

cycles.

4. The equation 2 is not suitable for alloy materials.

5. As the authors pointed out that the decomposition of LiPF₆ and reduction of FEC can be observed during cycle. It is quite strange that the cycle performance of o-dGZNFs is very stably.

Reviewer #3 (Remarks to the Author):

This paper reported atomic-scale combination of germanium-zinc distorted array for structural and electrochemical evolution. The authors investigated the atomic-scale combination of germanium-zinc distorted alloy exhibiting uncertain eutectic point in a phase diagram can imply a great potential to form atomically collaborated array via a simple fabrication method. Impressively, the features are confirmed through in situ analysis. More importantly, the anodes showed an improved electrochemical performance. Besides, this manuscript is well organized. Therefore, I recommend the acceptance of the manuscript to be published in Nature Communications after minor revisions according to the following aspects:

1. The instrument model in this experiment should be given.

2. In the Fig. 3d, the O-dGNFs shows a significant capacity gain from 50 cycles to 150 cycles when the current density changes to 2.0 C, why?

3. Some latest references should be cited, such as ACS Nano 12 (2018) 4993, Nano Energy 47 (2018) 503, Chemical Communications 53 (2017) 8360, ACS Applied Materials & Interfaces 9 (2017) 12400, Journal of Power Sources 363 (2017) 193, Journal of Power Sources 378 (2018) 66.

4. Writing is acceptable but could be further improved. There are many grammar mistakes and typos in the manuscript, such as "...they exhibits a high theoretical capacity...", "...as-prepared battery anodes remarkably highlights...". Please carefully proofread the manuscript.

5. Please make sure that your references titles are correct (such as reference 9).

<Responses to the reviewers' comments>

Manuscript ID: NCOMMS-18-14562

Title: Atomic-scale combination of germanium-zinc distorted array for structural and electrochemical evolution

Reviewer #1

Overall comment: *In this work, the authors report atomic-scale combination of germanium-zinc distorted alloy. These one-dimensional Ge-based materials with intermolecular incorporation of Zn into disordered Ge clusters can greatly enhance both structural integrity and electronic conductivity. Further, the unique features are clearly confirmed through in situ analysis. Besides, as-prepared battery anodes remarkably highlights outstanding rate capabilities and cycle retention with a capacity of 546 mAh g⁻¹ even after 1000 cycles. When assembled in a full cell, it notably facilitates considerable energy density during 400 cycles with 99.4% of average coulombic efficiency. We think it's a remarkable work, however there are still some problems must be figured out before it is published in the high-level journal. We suggest it can be published after major revision.*

Response: We thank the reviewer for the very positive comments. We also appreciated the reviewer's constructive comments, which are addressed upon revisions as detailed in the following. The revised and/or newly added parts are highlighted as red in the main manuscript and supplementary information.

Comment 1: *The existence of Zn can prevent a fatal sublimation of germanium/germanium oxide during gas-solid phase reduction reaction, however it can't be testified in*

Supplementary Fig. 1, the Ge content in O-dGNFs is still quite high, please explain. The sintering temperature is below the melting point of Ge, how to explain the sublimation of Ge occurs?

Response 1: We appreciate the reviewer's comment, which will strengthen our manuscript. GeO₂ can have transition states, such as Ge (g) or GeO (g), while it is reduced under hydrogen flow (*Trans. JIM.* **1972**, *13*, 39-44). These are unstable in the reaction, which result in substantial sublimation. Furthermore, as well-known properties of nanoparticles, the smaller of the particle size, the lower of the melting temperature (*Adv. Mater.* **2003**, *15*, 353-389). In our results seen in Supplementary Fig. 1, both of O-iGNFs and O-dGNFs have high Ge portion. Ge portion is almost similar even though carbon portion increases on reduction process. It is obvious that Ge portion relatively decreases on reduction process. That is, the Ge content in O-dGNFs is quite high because the as-spun nanofibers already have lots of Ge portion but the amounts are reduced after reduction reaction based on Fig. 2f in manuscript. On the other hand, ZnO cannot be reduced at 600 °C in hydrogen atmosphere, as shown in Supplementary Fig. 5. So, Ge that is chemically bonded with Zn has the role for the stabilization of Ge or GeO sublimation by the presence of Ge-O-Zn or Ge-Zn bonds, as demonstrated in Fig. 2f. In this reason, we can explain that sublimation of nanosized Ge and Ge nanofiber occurs in such lower temperature, below the melting point of Ge, and Zn can prevent the sublimation of Ge compounds. Based on these points, we have revised the manuscript, as shown below:

(Revised manuscript, page 3, line 35-36)

Although both O-dGNFs and O-dGZNFs have high portion of Ge, O-dGZNFs exhibit less Ge weight loss from O-iGZNFs compared with O-dGNFs.

Comment 2: *Why Zn is selected? If other metallic elements have similar effect?*

Response 2: Thanks to the reviewer for asking this great question. Based on phase diagram of binary metal system, elements that can form alloy with Ge are largely classified into two groups. First, for example, Cu or Fe metal can form alloys with various phases like Cu_3Ge and Fe_3Ge , and FeGe_2 because phase diagrams of Cu-Ge and Fe-Ge show wide ranges of specific alloying condition in Fig. RA-2 (*J. Alloy and Compd.* **2010**, 504, 159-165 and *J. Mat. Res.* 2006, 21, 174-184) as compared with Zn-Ge in Fig. RA-1 (*Thermochim. Acta* **1989**, 155, 227-240). In the case of alloying materials, we cannot expect atomic distribution of each element for structural novelty used in this system. Besides, when we synthesized Cu-Ge nanofiber using polymer solution that includes copper precursor instead of zinc precursor, we detected some crystallinity after calcination and Cu_3Ge alloy portion after reduction in Fig. RA-2c as compared with O-iGZNFs and O-dGZNFs. Further, when such alloying products are applied to anode materials, the resulting electrodes showed improved cycling stability compared to bare Ge electrode (*RSC Adv.* 2016, 6, 89176 and *Electrochim. Acta* 2017, 253, 552-529). However, even though these electrodes have similar electrical conductivity as O-dGZNFs (used in this study), they cannot reach long-term cycling stability and good rate capability compared with the O-dGZNFs. Second alloying types are comparable to the Zn-Ge system in Fig. RA-3 (*Bull. Alloy Phase Diagr.* **1988**, 9, 58-64, *Phys. Rev. B* **2009**, 80, 125414, *J. Phase Equilibria Diff.* **2012**, 33, 162, and *Opt. Mater. Express* **2014**, 3, 1385-1396). These phase diagrams suggest the possibility for homogeneous mixing such as example of Zn-Ge, Ag-Ge, Au-Ge, Sb-Ge, and Sn-Ge. However, Ag and Au metals are unfortunately expensive and the conditions are restrained owing to sensitive metal precursors. Besides, Sb has the difficulty in synthesizing via electrospinning method because the solubility of Sb precursor in

water and mixing condition with Ge precursors make them not suitable for this system. Sb-Ge can be a good candidate for structural novelty but it is not suitable for our synthetic system. As another example, Sn element features almost similar thermodynamic behavior when mixed with Ge and implies the feasibility of homogeneous structure with atom-level distribution, which is similar to Zn-Ge system. In experimental result in Fig. RA-3e, Sn-Ge nanofibers show overall amorphous phase after calcination with broad carbon position. Then, after reduction, those had been reduced to Ge, Sn, and GeSn with partial germanium oxide or tin oxide. As compared with Ge-Zn system, tin oxide has lower thermodynamics of reduction under hydrogen gas atmosphere than zinc oxide. So, tin should be emerged after reduction and some alloying with Ge. Furthermore, tin can melt down at given reduction temperature so Sn metal might nucleate each other showing both phase GeSn and Sn (*Phys. Chem. Chem. Phys.*, **2013**, 15, 11691-11695). Shortly, Sn and Ge binary system has the possibility to form Ge-Sn phase like Zn-Ge. However, intrinsic properties of Sn can bring out some different structural evolution from that of Ge-Zn system, O-dGZNFs, as shown in Fig. RA-3e. Sn-Ge nanofibers are also expected to show much worse electrochemical properties than O-dGZNFs. Sn element reacts with Li ion as many as Li_xSn ($0 < x < 4.4$) so that it can trigger larger volume expansion and then structural failure during long-term cycle test, whereas Li_yZn ($0 < y < 1$) provides limited volume expansion. Based on these reasons, we lastly chose Zn elements to make atomically distributed Zn-incorporated Ge nanofiber considering various parameters. Based on these points below, we have added sentences in the revised manuscript, with a figure in the supplementary information.

(Revised manuscript, page 3, line 22-30)

Among various kinds of metals, Zn was carefully selected as some metals (such as Cu and Fe) that form alloys with Ge, verified from X-ray diffraction (XRD) analysis before and after

reduction (Supplementary Fig. 1a). Moreover, other kinds of metals (such as Ag, Au, Sb, and Sn) can be combined with Ge similar to Zn, but other limitations are present. For Ag and Au, they are very expensive and synthetic conditions are difficult due to their sensitive precursors; for Sb, it is difficult to be mixed together with Ge in electrospinning solution; for Sn, it can be easily mixed with Ge and form various kinds of phases, from GeSn to partial Ge oxide and Sn oxide (Supplementary Fig. 1b). As a result, Zn was selected as the most optimal element to carry out further experiments to synthesize O-dGZNFs.

(Revised supplementary information, Fig. 1)

Supplementary Fig. 1. XRD patterns of (a) Ge-Cu and (b) Ge-Sn composite NFs after calcination and reduction process.

Figure RA-1. Phase diagram of Zn-Ge.

Figure RA-2. Phase diagram of **a** Cu-Ge and **b** Fe-Ge. **c** XRD patterns of Ge-Cu nanofibers after calcination and reduction in the same manner with O-dGZNFs except for copper precursor.

Figure RA-3. Phase diagram of **a** Ag-Ge, **b** Au-Ge, **c** Sb-Ge, and **d** Sn-Ge. **e** XRD patterns of Sn-Ge nanofibers after calcination and reduction in the same manner with O-dGZNFs except for tin precursor.

Comment 3: *The sintering temperature is higher than 450 °C, why there is still much carbon remained in the materials?*

Response 3: We appreciate the reviewer’s comment, which will strengthen our manuscript. As noted by the reviewer, it is necessary to explain how much carbon is remained in the materials. Based on the previous literature (*Mater. Res.* **2015**, *18*, 509-518), it can be observed that not all carbon is decomposed at 450 °C, and even at the temperature of 500 °C, some carbon remained. Especially, in the presence of metal precursor together with PVP, the decomposition of PVP is expected to be slower and occurs at higher temperature, as the crystallization of metal oxides formed from metal precursors also takes place.

Comment 4: *The statement and analysis of the bonding between Ge, Zn and O are ambiguous, please improve it. Some statement seems inconsistent with the data, for example the peak at 750 cm⁻¹ is almost sightless in O-dGZNFs, please modify or give some*

explanation.

Response 4: We appreciate the reviewer's comment, which will strengthen our manuscript. The dashed line in Fig. 2a was not exactly matched with real position. The shoulder exists in 750 cm^{-1} in Raman spectra. We re-aligned the peak position. We elaborated further in the manuscript to clarify some statements of the bonding between Ge, Zn, and O.

(Revised manuscript, Fig. 2a)

Fig. 2a Raman spectra of O-dGNFs and O-dGZNFs in metal region.

(Revised manuscript, page 4, line 24-26)

O-dGZNFs showed broad Ge-Ge bonds in the range of $280\text{-}305\text{ cm}^{-1}$ and independently exhibited oxygen defected and asymmetric Ge-Zn-O bonds at 750 and 777 cm^{-1} respectively as well as partial Zn-O bonds (437 cm^{-1}).

Comment 5: *How the O element content influence the electrochemical performance of O-dGZNFs, is there any experimental data to support it?*

Response 5: That is an excellent question. Oxygen contents can be controlled during the electrospinning process employing polymer solution with different Ge/Zn ratio. Depending on the ratio of Ge/Zn precursors, oxygen amounts are different after reduction process. When more Zn metal precursors are added in polymer solution, higher oxygen content is included in final sample, because ZnO bonding in the final product cannot be reduced at operating condition. In Fig. RB, each O-dGZNFs-1, 2, and 3 correspond to the Ge/Zn ratio of 10/1, 5/1, and 1/1 compared with O-dGZNFs (Ge/Zn = 2/1). At the Ge/Zn with above the ratio of 2/1, the cycle retention at 2 C-rate showed capacity fading as less oxygen contents are involved in nanofibers. Meanwhile, the O-dGZNFs-3 electrode (below Ge/Zn ratio of 2/1) shows very stable cyclability. However, the capacity is lower than other electrodes due to relatively lower Ge contents, which are main contributors for capacity. To summarize, oxygen contents are trade-offs between reversible capacity and cyclability. In case of high oxygen contents, the electrode can perform stable retention over few hundreds of cycles but showed lower reversible capacity. On the other hand, in case of low oxygen contents, the electrode can display high reversible capacity but the capacity cannot be retained for long cycles. Based on this point, we have revised the manuscript and supplementary information.

(Revised manuscript, page 6, line 12-18)

In addition, to examine the effect of oxygen content in the electrochemical performance, charge capacity retention tests (Supplementary Fig. 15) were carried out for O-dGZNFs with different Ge/Zn ratio, where higher portion of Zn leads to higher oxygen content. When the oxygen content is higher, it resulted in stable cycle retention owing to high ionic/electronic conductivity, but showed poor reversible capacity. On the other hand, when oxygen content

was lower, it resulted in higher initial capacity but capacity fading was more prominent.

(Revised supplementary information, Fig. 15)

Supplementary Fig. 15 Charge capacity retention of O-dGZNFs-1, 2, and 3 depending on different oxygen ratios (O-dGZNFs-1: Ge/Zn=10/1, O-dGZNFs-2: Ge/Zn=5/1, and O-dGZNFs-3:Ge/Zn=1/1).

Comment 6: Why the high temperature sintering treatment doesn't make the Ge and Zn become crystalline state?

Response 6: We are thankful to the reviewer for asking this question. It is known that GeO₂/carbon nanofiber remains as amorphous phase at 600 °C in air (*Chin. Chem. Lett.* **2016**, 27, 412-416). So, it is reasonable that O-iGNFs did not show some crystallinity after calcination. Similarly, O-iGZNFs, which have Zn-O-Ge bonding, can also remain as amorphous phase. Furthermore, to check the tendency of crystallization of our samples, we

conducted calcination process at 600 and 650 °C, respectively. As shown in Fig. RC, no specific crystal region was observed at 600 °C. However, when the temperature was increased to 650 °C, the crystalline particles were observed. These results are consistent with results of previous literatures and support structural properties of our samples. Based on this point, we have revised the manuscript and added a figure in the supplementary information, as shown below:

(Revised manuscript, page 4, line 5-7)

It should be noted that amorphous phase was maintained at 500 °C. When the sintering temperature increased to 600 and 650 °C, the crystalline structures of GeO₂ and Zn₂GeO₄ were developed (Supplementary Fig. 5).

(Revised supplementary information, Fig. 5)

Supplementary Fig. 5 XRD patterns of a O-iGNFs and b O-iGZNFs calcined at 600°C (black) and 650°C (red) in air.

Comment 7: *The annotation in Fig. 4c seems to have some mistake.*

Response 7: We appreciate the reviewer's comments, which correct our mistakes. We modified the annotation in Fig. 4c of the revised manuscript, as shown below:

(Revised manuscript, Fig. 4c)

Fig. 4c EXAFS spectra of O-dGNFs at various states.

Comment 8: *The Zn also suffers volume expansion during lithiation, why the volume fluctuation of O-dGZNFs is significantly smaller than that of O-dGNFs?*

Response 8: We appreciate the reviewer's comment and answer to this question will help to strengthen the manuscript. Zn also undergoes volume change inevitably due to the reaction with Li on alloying process (Li_xZn , $0 < x < 1$). Thus, O-dGZNFs electrode also features such volume expansion as shown in Supplementary Fig. 21. However, compared with that of Li_yGe ($0 < y < 4.4$), the ratio of Li_yZn is smaller than that of Li_yGe . It means that Zn-included nanofibers, O-dGZNFs, are relatively free from volume expansion compared with O-dGNFs,

composed of only Ge compounds. So, the results can be stemmed from Zn existence with less degree of volume expansion. In addition, Zn-O-Ge bonding produces Li₂O layer during lithiation process. The Li₂O layer can also act as buffering layer which can accommodate the volume expansion of Zn/Ge compounds. Based on two reasons above, the volume fluctuation of O-dGZNFs was significantly smaller than that of O-dGNFs. Based on this point, we have revised the manuscript, as shown below:

(Revised manuscript, page 8, line 15-17)

Based on these results, Zn helped to alleviate the volume expansion, which can be ascribed to better structural integrity arising from formation of Ge-Zn-O and lower theoretical volume changes of Zn.

Comment 9: *Why the conductance of O-dGZNF in state iii was still 31 times higher than that of O-dGNF even after complete lithiation of both NFs? In this state the Zn is transformed to Li-Zn.*

Response 9: Thanks to the reviewer for this great comment. Intrinsic properties, in aspect of electrical property, are at first different for all elements. That is, Ge and Zn have different electronic conductivities ($2 \times 10^3 \text{ S m}^{-1}$ for Ge, $1.7 \times 10^7 \text{ S m}^{-1}$ for Zn in bulk state). This means that Zn has intrinsically much higher conductivity than Ge. Secondly, lithiated Si (with similar property as lithiated Ge) particle shows enhanced electrical properties as described in the example of Li_xSi compared to bare Si (*ACS Nano* **2014**, 8, 11816-11826). Therefore, as the lithiation proceeds more, the electrical conductivity also increases. With two reasons mentioned above, the conductance of O-dGZNFs features such higher value than that of O-

dGNFs after complete lithiation. Based on this point, we have revised the manuscript:

(Revised manuscript, page 8, line 35-37)

Furthermore, the conductance of O-dGZNF in state iii was still 31 times higher than that of O-dGNF even after complete lithiation of both NFs – this is attributed to the higher intrinsic conductivity of Zn ($1.7 \times 10^7 \text{ S m}^{-1}$) compared with that of Ge ($2 \times 10^3 \text{ S m}^{-1}$) in bulk and lithiated Ge also exhibits enhanced electrical properties.

Comment 10: *As the authors state, the Li_2O and Li_2CO_3 have important impact to the ionic conductivity and conductance of the materials, if some quantitative experimental data can be supplied to support the conclusion?*

Response 10: The reviewer raised a great question. It is known that Li_2O and Li_2CO_3 have the role of ion conductors. Thus, the ionic conductivity eventually increases on cycling because the amount of Li_2O and Li_2CO_3 increases by electrochemical reaction with O-dGNFs or O-dGZNFs in the electrode. For O-dGNFs and O-dGZNFs electrode after 50 cycles (O-dGNFs-50th/O-dGZNFs-50th), as compared with those of one-cycled and 10 cycled electrodes (O-dGNFs-1st/O-dGZNFs-1st and O-dGNFs-10th/O-dGZNFs-10th), the ion diffusion coefficients of both electrodes showed improvement with cycling numbers. Based on this point, we have newly added a graph summarizing the ion diffusion coefficients of O-dGNFs and O-dGZNFs after the 1st, 10th, and 50th cycle, calculated the ion diffusion coefficients, and revised the manuscript, as shown below:

(Revised manuscript, page 7, line 9-11)

When the Li ion diffusion coefficients were further compared after the 1st, 10th, and 50th cycle, they gradually increased, which are proportional to the increased amount of lithium composites (Supplementary Fig. 17 and Supplementary Table S1).

(Revised supplementary information, Table S1)

Supplementary Table S1. Comparison of Li ion diffusion coefficients calculated from Fig. 3f.

Sample	Ion diffusion coefficient ($\text{cm}^{-2} \text{s}^{-1}$)	Sample	Ion diffusion coefficient ($\text{cm}^{-2} \text{s}^{-1}$)
O-dGNFs-1st	8.958×10^{-8}	O-dGZNFs-1st	2.702×10^{-7}
O-dGNFs-10th	1.092×10^{-7}	O-dGZNFs-10th	2.726×10^{-7}
O-dGNFs-50th	1.257×10^{-7}	O-dGZNFs-50th	3.462×10^{-7}

(Revised supplementary information, Fig. 17)

Supplementary Fig. 17 Comparison of Li ion diffusion coefficients for O-dGNFs and O-dGZNFs depending on the different cycles.

Reviewer #2

Overall comment: *This work reported the preparing of Ge-Zn nanofiber and studied its lithium storage properties. It seems that the as-prepared battery anodes shows improved lithium storage properties, including rate capabilities and cycle performance. This manuscript is good at materials characterization but the results analysis is not sufficient. I don't recommend it is accepted. There are several important points I wish to bring up.*

Response: We appreciate the reviewer to give helpful comments, which can strengthen our manuscript. Usually, numerous researches have been devoted to fabricating various kinds of Ge-based nanostructures, in the form of core-shell, with coating layers, and/or alloy with certain stoichiometry. But, systematic and in-depth study on the feasible method to minimize the sublimation of Ge upon heat treatment while optimizing Ge-based anodes with structural evolution of new chemical bonding has yet been presented. In terms of methods, we have successfully fabricated O-dGZNFs with minimal loss of Ge by introducing Zn to prevent the sublimation of Ge by electrospinning and subsequent heat treatments, which are simple and possible for mass production. In aspect of structural evolution, our suggested electrode features abnormal atomic distribution. Such novel discovery is attributed to the unique Ge/Zn binary phase diagram, which does not form alloys of Ge_xZn_y . To further probe into the exact properties and the reasons that attribute to such structural evolution, X-ray based instrument including XRD, XPS, and XAFS in atomic level were used. What is more, our study has further delved into how such structural evolution of Ge induced by the introduction of Zn can significantly enhance the electrochemical performances when used as anodes in lithium-ion batteries. Both *in situ* TEM/EIS and *ex situ* XAFS analyses were carefully carried out to observe morphological and electrochemical behavior while lithium ions were

inserting/extracting. Various tools systematically support how the structure had been evolved in synthetic process and how as-prepared sample can possess such enhanced structural/electrochemical stability as the anode material. To briefly summarize, the main points of novelty and strength in this paper are as follows: 1) This work firstly suggests a feasible strategy to fabricate Ge-based anodes upon reduction without significant weight loss of Ge due to sublimation. 2) We systematically investigate the unusual structural evolution of Ge when Zn is introduced. 3) We further probes into the reaction mechanism of newly synthesized O-dGZNFs upon lithium insertion/extraction, and suggests their feasibility as full-cell anodes. In the preparation of this revised submission, all authors tried our best to carefully address all the valuable comments and concerns from reviewers. The revised and/or newly added parts are highlighted as red letters in the main manuscript and supplementary information.

Comment 1: *Are there carbon and nitrogen element in Ge-Zn nanofiber? Carbon and nitrogen element can be seen in Fig. 1e. This is very important.*

Response 1: That is an excellent question. Because we used PVP polymer to fabricate one-dimensional nanofiber, carbon and nitrogen elements have to be included in our samples. So, we confirmed the distribution of carbon and nitrogen atoms **in Fig. 1e**. Further, we specified the weight percent of each carbon and nitrogen using EDS/EA analysis as shown in Supplementary Fig. 2. These data can be helpful to understand the existence of carbon and nitrogen in nanofibers.

Comment 2: *The carbon derived from PVP also improves the electronic conductivity.*

Response 2: That is a great point. As noted by the reviewer, it is important to know the contribution of carbon derived from PVP to the overall conductivity. To start with, it is important to note that both O-dGNF and O-dGZNF have carbon remained in the sample, which is presumably derived from PVP. In the previous literature (*Chem. Eng. J.* **2016**, *304*, 511-517), it is claimed that carbon derived from PVP improves the electronic conductivity. As commented by the reviewer, we agree that the PVP derived carbon also improves the electronic conductivity. Nevertheless, since both O-dGNFs and O-dGZNFs have small amount of carbon remained from PVP, the effect of carbon derived from PVP on the electronic conductivity is negligible. This can be another great research topic that can be delved in near future.

Comment 3: *Why the capacitive contribution of o-dGNFs is higher than o-dGZNFs? Why the capacitive contribution was increased after 50th cycles? Please check the particle size before and after 50th cycles.*

Response 3: We thank the reviewer to raise this great point. At first, both electrodes have higher capacitive currents rather than faradaic currents due to quite high surface area of one-dimensional nanofibers. In contrast, after 50 cycles, metallic bonds are developed in shape of Ge-Ge or Ge-Zn in the electrode with Li_2O and Li_2CO_3 formation. It means that these bonds can increase the kinetics of charge transfer under same condition of electrode before cycling. For this reason, cycled electrodes displayed higher faradaic current. Additionally, Ge-Zn existence in O-dGZNFs-50th can improve electronic conductivity, followed by more faradaic contribution, compared with the presence of only Ge-Ge bonds in O-dGNFs-50th.

In aspect of particle size, diameters of pristine nanofibers of O-dGNF and O-dGZNF are measured around 62 nm and 65 nm, respectively. The volume of electrodes is inevitably

expanded during cycling. So, there are some changes of nanofibers diameters before and after cycling. The nanofibers after 50 cycles showed expanded diameters of about 128 nm (O-dGNF) and 102 nm (O-dGZNF) in particle level as shown in Supplementary Fig. 22. Further, in electrode level, 300-cycled electrodes of O-dGNFs and O-dGZNFs brought out the volume change of 211% and 100%, respectively, as shown in Supplementary Fig. 21. Nevertheless, both scales showed less volume change rather than theoretical value. Also, alloying stoichiometry between Zn and Ge with Li is different - Li_xGe and Li_yZn ($0 < x < 4.4$, $0 < y < 1$). Therefore, the O-dGZNFs (Zn-included GNFs) showed less volume change compared with O-dGNFs. Based on these two reasons, the O-dGZNFs electrode exhibited smaller swelling ration rather than O-dGNFs. Reminded of the reviewer's comment, we have indicated the part related to this comment in the revised manuscript:

(Revised manuscript, page 6, line 27-36)

Capacitive current is related to surface reaction at the interface between electrolyte and electrode while faradaic current is concerned with charge transfer redox reaction of the electrode. In other words, total current in CV measurements consists of faradaic and capacitive current. In this viewpoint, Fig. 3e shows that capacitive current of both electrodes was dominant in all scan rates from 0.4 to 1.0 mV s^{-1} due to high surface area of one-dimensional structure (capacitive current: 65.8% of O-dGNFs and 78.5% of O-dGZNFs). Especially, O-dGZNFs electrode, which has higher BET surface, showed surface reaction corresponding to higher capacitive current than O-dGNFs electrode. However, the dominant current was reversed to the faradaic current after 50th cycle for both electrodes, which can be ascribed to formed lithium composites such as Li_2O and Li_2CO_3 as well as developed metallic bonds (Ge-Ge or Ge-Zn). Especially, Zn has better intrinsic electronic conductivity than Ge,

leading to higher charge transfer rate and more faradaic currents in O-dGZNFs.

Comment 4: *The equation 2 is not suitable for alloy materials.*

Response 4: We appreciate the reviewer's good suggestion. Different from other alloy materials such as Cu_3Ge and Fe_3Ge (*J. Mat. Sci.* **1998**, 33, 5405-5414 and *J. Alloy and Compd.* **2010**, 504, 159-165), Ge and Zn atoms do not have specific alloying stoichiometries based on phase diagram (*Bull. Alloy Phase Diagr.* **1985**, 6, 540-543). So, we think that our materials, which have atomic mixture, are not classified into alloying material. That is why we used equation 2, Randle-Sevcik equation for the calculation of diffusion coefficients.

Comment 5: *As the authors pointed out that the decomposition of LiPF_6 and reduction of FEC can be observed during cycle. It is quite strange that the cycle performance of o-dGZNFs is very stably.*

Response 5: We appreciate the great point of the reviewer. Usually, LiPF_6 and FEC are decomposed at around 1 V to form SEI layer on the anode surface. By-products of LiPF_6 and FEC decomposition are LiF , $\text{Li}_x\text{PO}_y\text{F}_z$, and F-related carbon group as shown in Supplementary Fig. 14. F-related organic carbon group especially has flexibility to accommodate volume change on alloying reaction with Li. So, the by-product of LiPF_6 and FEC influences cycle retention. However, stable cycle performance is not only determined by mechanical aspect, such as the decomposition of LiPF_6 and reduction of FEC. As another factor, O-dGZNFs electrode possesses Ge-O-Zn binding in NFs unlike O-dGNFs electrode. The Zn portion in electrode can perform high electronic conductivity and this factor directly affects to high current density stability. That is why O-dGZNFs electrode at 2.0 C-rate shows

quite stable electrochemical results (Fig. 3d) while O-dGNFs eventually lose their reversible capacity upon cycling due to poor electronic conductivity even though they have stable SEI layer on the surface of electrode.

Reviewer #3

Overall comment: *This paper reported atomic-scale combination of germanium-zinc distorted array for structural and electrochemical evolution. The authors investigated the atomic-scale combination of germanium-zinc distorted alloy exhibiting uncertain eutectic point in a phase diagram can imply a great potential to form atomically collaborated array via a simple fabrication method. Impressively, the features are confirmed through in situ analysis. More importantly, the anodes showed an improved electrochemical performance. Besides, this manuscript is well organized. Therefore, I recommend the acceptance of the manuscript to be published in Nature Communications after minor revisions according to the following aspects:*

Response: We appreciate the reviewer's positive comment of our work. We also thank the reviewer for the constructive comments and suggestions, which are addressed in the following.

Comment 1: *The instrument model in this experiment should be given.*

Response 1: This is an excellent point. We added the instrument models for electrospinning, calcination, and reduction process in revised manuscript, as shown below:

(Revised manuscript, page 10, line 27 - page 11, line 3)

Synthesis of O-iGNFs and O-iGZNFs. O-iGNFs and O-iGZNFs were prepared via electrospinning and subsequent calcination step. For the preparation of O-iGNFs, 0.4 g of GeO₂ was dissolved in 50 mL of de-ionized (DI) water at 90 °C. Then, 7.5 g of PVP was added into the solution and stirred at 500 rpm for 6 h. Then, the electrospinning solution was loaded into the syringe, where the electrospinning process took place using electrospinning tool (Machine 1 Type, NanoNC). The electrospinning was conducted with the following conditions: flow rate of 0.5 mL h⁻¹, an applied voltage of 16.0 kV, and a distance of 15 cm between the tip of syringe and current collector using 25 gauge needle. As-spun NFs were calcined at 250 °C for 1 h and 500 °C for 2 h at a ramping rate of 5 °C min⁻¹ in box furnace (Vulcan 3-550, NAvtech). Meanwhile, for the preparation of O-iGZNFs, a solution mixture with GeO₂ and Zn (NO₃)₂·6H₂O (98%), which results in specific amounts shown in Supplementary Fig. 2b, was added to the electrospinning solution with the identical electrospinning conditions and subsequent calcination step. As a result, O-iGNFs and O-iGZNFs were successfully synthesized.

Synthesis of O-dGNFs and O-dGZNFs. As-synthesized O-iGNFs and O-iGZNFs underwent reduction process in quartz furnace (OTF-1200X-II, MTI corporation) filled by argon (Ar). In this process, furnace under Ar was heated to 600°C at a ramping rate of 5 °C min⁻¹ and once the temperature reached an expected value, the atmosphere was changed to Ar and hydrogen gas mixture (Ar/H₂ (96/4, v/v)) and maintained for 1 h. Afterward, the furnace was filled again by Ar instead of Ar/H₂ and cooled down spontaneously to room temperature.

Comment 2: *In the Fig. 3d, the O-dGNFs shows a significant capacity gain from 50 cycles to 150 cycles when the current density changes to 2.0 C, why?*

Response 2: That is a good question. Nanosized materials usually need the activation step to attain maximum capacity in the electrode when the current density is changed. Especially, O-dGNFs have inferior electronic conductivity compared with O-dGZNFs as mentioned in the manuscript. So, the O-dGNFs require longer time to activate their own capacity at a given current density. That is why the O-dGNFs electrode seems to have capacity gain for few ten cycles. Based on this point, we have revised the manuscript, as shown below:

(Revised manuscript, page 6, line 11-12 and 18-21)

O-dGZNFs after 50th cycle displayed outstanding reversible capacity and cycle retention until 350 cycles at even 2.0 C-rate with almost 100% capacity retention (0.05% capacity decay per each cycle).

In contrast, O-dGNFs require activation step to attain optimal capacity, showing capacity gain during few ten cycles. However, this electrode subsequently suffered from fatal capacity decay with only 49.5% retention (corresponding 0.3% capacity decay per each cycle) due to the low electronic conductivity and poor sustainability to respond to current changes.

Comment 3: *Some latest references should be cited, such as ACS Nano 12 (2018) 4993, Nano Energy 47 (2018) 503, Chemical Communications 53 (2017) 8360, ACS Applied Materials & Interfaces 9 (2017) 12400, Journal of Power Sources 363 (2017) 193, Journal of Power Sources 378 (2018) 66.*

Response 3: We are thankful to the reviewer for suggesting these suitable references. We added the suggested references in the revised manuscript (from 5 to 10 in the reference lists), as shown below:

(Revised manuscript, Ref. 5-10)

5. An, Y. *et al.* Commercial expanded graphite as a low-cost, long-cycling life anode for potassium-ion batteries with conventional carbonate electrolyte. *J. Power Sources* **378**, 66-72 (2018).
6. An, Y. *et al.* Lithium metal protection enabled by in-situ olefin polymerization for high-performance secondary lithium sulfur batteries. *J. Power Sources* **363**, 193-198 (2017).
7. An, Y. *et al.* Vacuum distillation derived 3D porous current collector for stable lithium-metal batteries. *Nano Energy* **47**, 503-511 (2018).
8. An, Y. *et al.* Green, scalable, and controllable fabrication of nanoporous silicon from commercial alloy precursors for high-energy lithium-ion batteries. *ACS nano* **12**, 4993-5002 (2018).
9. An, Y. *et al.* Ultrafine TiO₂ confined in porous-nitrogen-doped carbon from metal-organic frameworks for high-performance lithium sulfur batteries. *ACS Appl. Mater. Interfaces* **9**, 12400-12407 (2017).
10. An, Y. *et al.* A titanium-based metal-organic framework as an ultralong cycle-life anode for PIBs. *Chem. Commun.* **53**, 8360 (2017).

Comment 4: Writing is acceptable but could be further improved. There are many grammar mistakes and typos in the manuscript, such as "...they exhibits a high theoretical capacity...", "...as-prepared battery anodes remarkably highlights...". Please carefully proofread the manuscript.

Response 4: We appreciate the reviewer's suggestion. We modified missing points in the revised manuscript, as shown below:

(Revised manuscript, page 5, line 21-23)

Ge anodes feature intrinsic inferior electronic conductivity and undergo a huge volume change during cycling in LIBs, triggering poor electrochemical performance even though they exhibit a high theoretical capacity.

(Revised manuscript, page 2, line 9-11)

Besides, as-prepared battery anodes remarkably highlight outstanding rate capabilities (capacity retention of ~50% at 20 C compared to 0.2 C-rate) and cycle retention (73% at 3.0 C-rate) with a capacity of 546 mAh g⁻¹ even after 1000 cycles.

Comment 5: *Please make sure that your references titles are correct (such as reference 9).*

Response 5: We thank the reviewer for identifying this editorial deficiency. We have fixed these deficiencies in the revised version. We have fixed the reference 9 as shown below:

(Revised manuscript, references)

11. Choi, S., Kim, J., Choi, N. S., Kim, M. G. & Park, S. Cost-effective scalable synthesis of mesoporous germanium particles via a redox-transmetalation reaction for high-performance energy storage devices. *ACS Nano* **9**, 2203-2212 (2015).

15. Hwang, J. *et al.* Mesoporous Ge/GeO₂/carbon lithium-ion battery anodes with high capacity and high reversibility. *ACS Nano* **9**, 5299-5309 (2015).

28. Xie, Y. H. *et al.* Quinary wurtzite Zn-Ga-Ge-N-O solid solutions and their photocatalytic properties under visible light irradiation. *Sci. Rep.* **6**, 19060 (2016).
36. Echeverria, J., Falceto, A. & Alvarez, S. Zinc-zinc double bonds: a theoretical study. *Angew. Chem. Int. Ed.* **56**, 10151-10155 (2017).
37. Zhao, J., Yang, L. J., McLeod, J. A. & Liu, L. J. Reduced GeO₂ nanoparticles: electronic structure of a nominal GeO_x complex and its stability under H₂ annealing. *Sci. Rep.* **5**, 17779 (2015).
49. Liu, X. H. *et al.* Reversible nanopore formation in Ge nanowires during lithiation-delithiation cycling: an in situ transmission electron microscopy study. *Nano. Lett.* **11**, 3991-3997 (2011).

REVIEWERS' COMMENTS:

Reviewer #1 (Remarks to the Author):

The authors have answered the questions and modified the manuscript carefully. We think they have figured out the questions well and we recommend it to be published in Nature Communications.

Reviewer #2 (Remarks to the Author):

This article describes the fabrication of oxygen defected and intermolecular distributed Ge-Zn nanofibers via a facile electrospinning method and subsequent thermal treatment. Such intermolecular interaction of Ge-Zn in NFs not only resulted in enhanced structural integrity but also enabled faster electron and ionic transport, thus leading to excellent lithium storage performance. However, despite a lot of tests and characterization have been presented, the ideas and methods of this article are not sufficiently innovative. What's more, some relevant data is not rigorous enough, such as the HAADF-STEM image in Fig. 1e. Overall, I think this article couldn't reach the high level of Nature communication. Following are the revised suggestion.

- 1) Corresponding SEM and TEM images with high magnification should be provided, as well as the HAADF-STEM image with higher resolution.
- 2) The capacity curves of the O-dGZNFs anode for featured cycles should be optimized.
- 3) The composition and charge-discharge process of the O-dGNFs and O-dGZNFs electrodes should be provided.

Reviewer #3 (Remarks to the Author):

It is well revised and can be accepted now.

<Responses to the reviewers' comments>

Manuscript ID: NCOMMS-18-14562

Title: Atomic-scale combination of germanium-zinc nanofibers for structural and electrochemical evolution

Reviewer #2

Overall comment: *This article describes the fabrication of oxygen defected and intermolecular distributed Ge-Zn nanofibers via a facile electrospinning method and subsequent thermal treatment. Such intermolecular interaction of Ge-Zn in NFs not only resulted in enhanced structural integrity but also enabled faster electron and ionic transport, thus leading to excellent lithium storage performance. However, despite a lot of tests and characterization have been presented, the ideas and methods of this article are not sufficiently innovative. What's more, some relevant data is not rigorous enough, such as the HAADF-STEM image in Fig. 1e. Overall, I think this article couldn't reach the high level of Nature communication. Following are the revised suggestion.*

Response: We appreciate the reviewer's critical comments, which will strengthen our manuscript. Alloy compounds are typically formed with specific molecular ratio in shape of A_xB_y based on phase diagram. Among numerous metals in periodic table, several metals do not organize alloy-type materials as specific forms like germanium-zinc. It means that bimetallics can mix each other without any chemical formula as mentioned above (e.g. A_xB_y). With this intrinsic behavior, we focused on atomic-level configuration of germanium and zinc metals in one-dimensional structure suitable for lithium-ion batteries. So, we can build structural system efficiently in atomic scale for realizing each critical role of metals that germanium mainly displays high reversible capacity and zinc provides considerable

electronic conductivity, respectively. In this regard, our synthetic strategy can be distinguished from general synthesis of alloying material. Further, Electrochemical/structural behavior of as-prepared samples is clearly demonstrated through *in situ/ex situ* analysis techniques with various instruments. All authors are devoted to reflecting reviewer's meaningful comments.

Comment 1: *Corresponding SEM and TEM images with high magnification should be provided, as well as the HAADF-STEM image with higher resolution.*

Response 1: We thank the reviewer's suggestion to elucidate further on our results. It is important to provide high quality electron microscopy images in order to clear up morphological structure of samples and accurate crystal structure to confirm atomic-level configuration. We tried to re-obtain high-magnified and high-resolution electron microscope images. Now we provide improved images to demonstrate clear morphological structure and atomic distribution in nanofibers in Figure RA and B. Then, we modified and added the images in Fig. 1. Please see the revised manuscript.

Figure RA. High-magnified TEM image of O-dGZNFs. Scale bar: 5 nm.

Figure RB. High-magnified STEM image and element mapping results. Scale bar: 50nm

Fig. 1 Morphological structure evolution. (a) Schematic illustration for whole synthesis process. TEM images inset of (a) correspond to O-iGNZFs and O-dGNZFs, respectively. (b) SEM image as-spun NFs. HR-TEM image and SAED patterns of (c) O-iGNZFs and (d) O-dGNZFs. (e) HAADF-STEM mapping of O-dGNZFs. Scale bars: (a) 500 nm, (b) 10 μm , (c,d) 5 nm, 5 1/nm, and (e) 50 nm.

Comment 2: *The capacity curves of the O-dGZNFs anode for featured cycles should be optimized.*

Response 2: We appreciate the reviewer's great point to elucidate electrochemical properties. Our sample experienced two kinds of steps, which are conversion and alloying/dealloying reaction upon charge/discharge process. For structural comparison, ex situ XAFS and in situ TEM deliver fine structural evolution for each as-prepared sample (O-dGNFs and O-dGZNFs) as well as cycled electrodes to investigate metallic bonds and imperfect oxide reformation in the electrodes. As a result, we conclude that oxide form, Ge-Zn-O, disappears after 50 cycles and metallic Ge-Zn or Ge-Ge bonds with Li₂O compounds had been emerged, suitable for high power lithium ion batteries. Further, we obtained enhanced electronic/ionic conductivity. Here, to clearly optimize and confirm the structural evolution, we added charge/discharge curve at various cycle states. As shown in Figure RC, initial 10 cycles have slight plateau, attributed to conversion reaction for the oxide reformation, related to equation (R1). But, the plateau is eventually reduced and finally the shape is changed to linear slope at followed cycles while alloying reaction parts still remain reversibly, assigned to equation (R2). Thus, we optimized the structural evolution by observing different two steps in galvanostatic charge/discharge voltage profile at various cycle states

Figure RC. Galvanostatic charge/discharge voltage profile for featured cycles. Oxide reformation behavior (Ge-Zn-O), occurred at around 1 V vs. Li/Li⁺, is not reversible well on cycling owing to inferior catalytic ability of active materials. Instead, Ge-Zn or Ge-Ge metallic bonds were eventually developed at lower 1 V vs. Li/Li⁺ with Li₂O compounds.

Comment 3: *The composition and charge-discharge process of the O-dGNFs and O-dGZNFs electrodes should be provided.*

Response 3: We appreciate the reviewer's great point, which will strengthen our manuscript. As shown in Fig. 3 and as-prepared response 2, both electrodes gradually lost the contribution of oxide reformation at around 1 V vs. Li/Li⁺. For comparison of electrodes after charge-discharge process, XPS results in Fig. 4 and Supplementary Figure 14 show binding energy shift, which means that Ge-Zn-O is converted into Ge-Zn well and that Li₂O portion is increased during charge-discharge process of electrodes. In case of O-dGNFs, they exhibit Ge⁰ emergence and Ge-O disappearance in Fig. 4c and Supplementary Figure 14e. Existence of Li₂O is also demonstrated through ion diffusion coefficient changes because Li₂O components can enhance ionic conductivity of electrode in Supplementary Figure 17. Shortly,

the composition of electrode firstly consists of Ge-Ge, Ge-Zn and Ge-Zn-O and they are eventually converted into Ge-Ge, Ge-Zn, and Li₂O at cycled electrode. Moreover, related to Fig. 3a, O-dGNFs electrode has two-step reactions like equation (R3) and (R4) at initial charge-discharge process but, equation (R4) is gradually dominant at the following cycles. Meanwhile, in accordance with Fig. 3b, O-dGZNF electrode has similar behavior, assigned to the equation (R1) and (R2). Likewise, equation (R2) is finally dominant, forming Li₂O compounds permanently.